# Unleashing Diffusion Transformers for Visual Correspondence by Modulating Massive Activations

Chaofan Gan[1,2]    Yuanpeng Tu[3]    Xi Chen[3]    Tieyuan Chen[1]
Yuxi Li[1]    Mehrtash Harandi[2]    Weiyao Lin[1]*
[1]Shanghai Jiao Tong University    [2]Monash University
[2]The University of Hong Kong
{ganchaofan, tieyuanchen, lyxok1, wylin}@sjtu.edu.cn
{chaofan.gan, mehrtash.harandi}@monash.edu
{tjtuyuanpeng, xichen.csai}@gmail.com

## Abstract

Pre-trained stable diffusion models (SD) have shown great advances in visual correspondence. In this paper, we investigate the capabilities of Diffusion Transformers (DiTs) for accurate dense correspondence. Distinct from SD, DiTs exhibit a critical phenomenon in which very few feature activations exhibit significantly larger values than others, known as *massive activations*, leading to uninformative representations and significant performance degradation for DiTs. The massive activations consistently concentrate at very few fixed dimensions across all image patch tokens, holding little local information. We analyze these dimension-concentrated massive activations and uncover that their concentration is inherently linked to the Adaptive Layer Normalization (AdaLN) in DiTs. Building on these findings, we propose the **Di**ffusion **T**ransformer **F**eature (DiTF), a training-free AdaLN-based framework that extracts semantically discriminative features from DiTs. Specifically, DiTF leverages AdaLN to adaptively localize and normalize massive activations through channel-wise modulation. Furthermore, a channel discard strategy is introduced to mitigate the adverse effects of massive activations. Experimental results demonstrate that our DiTF outperforms both DINO and SD-based models and establishes a new state-of-the-art performance for DiTs in different visual correspondence tasks (e.g., with +9.4% on Spair-71k and +4.4% on AP-10K-C.S.).

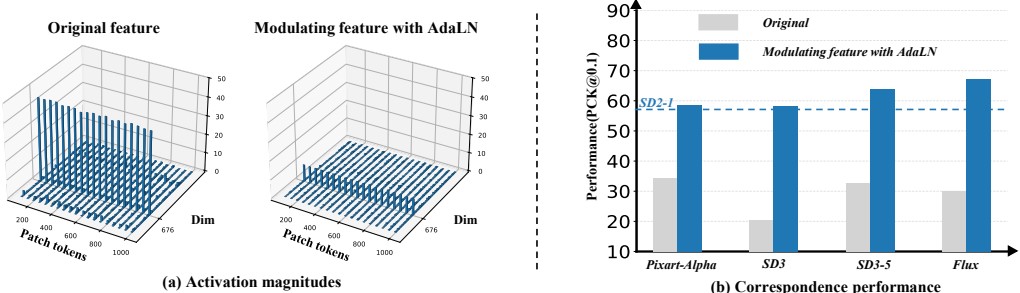

Figure 1: **AdaLN enhances DiT features by mitigating massive activations.** (a) Original DiT features show concentrated massive activations. (b) Semantic correspondence performance using different features. Original DiT features yield poor performance due to the presence of massive activations. By modulating these activations, AdaLN significantly boosts correspondence performance.

---

*Corresponding Author.

39th Conference on Neural Information Processing Systems (NeurIPS 2025).

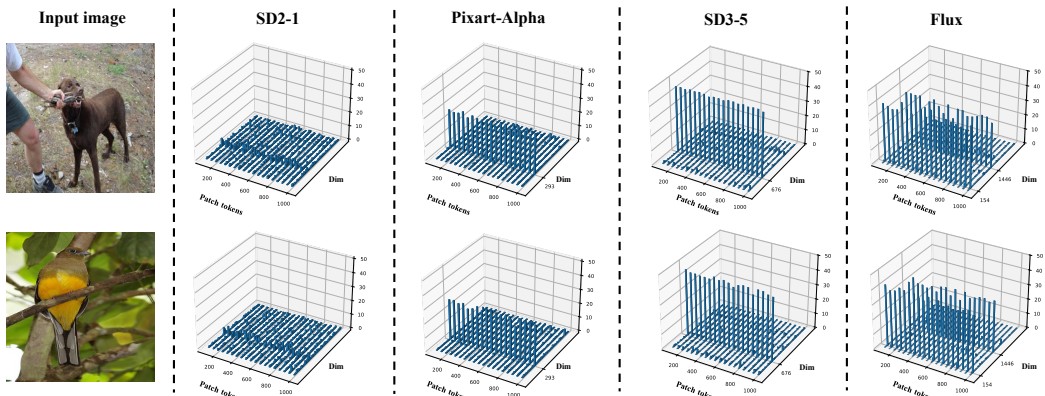

Figure 2: **Massive activations in Diffusion Transformers (DiTs).** We visualize the activation magnitudes (z-axis) of features across various diffusion models. Unlike the Stable Diffusion (SD2-1) model, all DiTs exhibit a distinctive phenomenon where very few feature activations show significantly higher activation values, more than 100 times larger than others. We refer to this phenomenon as *massive activations*, which has also been observed in large language models (LLMs) [20].

# 1   Introduction

Detecting dense correspondence between images plays an important role in various vision tasks such as segmentation [1, 2, 3, 4], image editing [5, 6, 7], and point tracking [8, 9, 10, 11]. Various works [12, 13, 14] utilize pre-trained self-supervised models like DINOv2 [12] to extract image features, which are then adapted for precise semantic correspondence. With the advent of text-to-image (T2I) generative models [15, 16], recent studies [17, 18, 14] have demonstrated the powerful capability of pre-trained SD [15] as an effective feature extractor [14, 18, 19] for visual correspondence.

Compared with SD, recent DiTs (e.g., Pixart-alpha [21], SD3 [22], and Flux [23]) demonstrate great superiority in scalability, robustness, model efficiency, and generation quality. While pre-trained SD models have been successfully leveraged as feature extractors, the potential of DiTs for visual perception tasks remains largely unexplored. To bridge this gap, we take the first step to investigate the feasibility of using pre-trained DiTs as feature extractors for visual correspondence tasks.

However, when directly extracting the original features for representation learning, as done in SD, DiTs exhibit disappointing performance for visual perception tasks, as shown in Figure 1. To better understand this discrepancy, we analyze the statistical properties of features extracted from both SD and DiTs (see Figure 2). Notably, we observe that DiT feature maps exhibit massive activations, in contrast to the smoother distribution observed in SD2.1 [15]. These massive activations share a few *fixed dominant* dimensions, resulting in a *high degree of directional similarity* for feature vectors in the latent space [24]. This property makes it difficult to distinguish spatial feature vectors using cosine similarity, leading to semantically uninformative and indiscriminate representations.

In this work, we aim to deepen the understanding of massive activations for representation learning, thereby enabling the extraction of semantically discriminative representations from DiTs. We begin by investigating the spatial and dimensional properties of these activations and find that they consistently emerge in a small set of fixed feature dimensions across all image tokens, which carry limited local information that is crucial for our task. Further analysis reveals an intriguing connection between massive activations and the AdaLN in DiTs, where the concentrated dimensions of massive activations align with the residual scaling factors produced by AdaLN. To further understand this phenomenon, we investigate the role of AdaLN in modulating DiT features. Our findings reveal that the built-in AdaLN effectively localizes massive activations and performs semantically meaningful channel-wise modulation to suppress them, thereby enhancing the semantic richness and discriminative capacity of DiT features.

Based on these findings, we propose a training-free AdaLN-based framework: **Di**ffusion **T**ransformer **F**eature (DiTF), to extract structural and semantic features from DiTs. Specifically, DiTF leverages the built-in AdaLN in DiTs to adaptively normalize massive activations through channel-wise modulation. In addition, a channel discard strategy is introduced to further mitigate the adverse effects of massive activations on representation learning. Our main contributions are summarized as follows.

- We identify and characterize massive activations in DiTs and find them consistently concentrated in a small number of fixed feature dimensions across all spatial tokens.
- We analyze the source of these dimensionally concentrated massive activations and uncover their strong connection to the AdaLN layer in DiTs.
- We propose a training-free AdaLN-based framework, **Di**ffusion **T**ransformer **F**eature (DiTF), for extracting semantically discriminative representations from DiTs.
- Our framework outperforms both DINO and SD-based models, establishing a new state-of-the-art for Diffusion Transformers on visual correspondence tasks.

## 2 Related Work

**Visual Correspondence.** Visual correspondence aims to establish dense correspondence between two different images, which plays a crucial role in various visual tasks such as segmentation [1, 2, 3, 4], image editing [5, 6, 7], and point tracking [8, 9, 10, 11]. Early methods like HOG [25], and SIFT [26] design hand-crafted features with scale-invariant key-points and then establish image correspondences. Various works attempt to learn image correspondence in a supervised learning regime. However, the supervised-based methods [27, 28, 29, 30, 31] depend on ground-truth correspondence annotations, which are challenging to apply to datasets lacking precise correspondence annotations. To handle this challenge, some studies have employed DINO [12, 32] and Generative Adversarial Networks (GAN) [33] to extract visual features for correspondence. More recently, [34, 14, 19, 35] have demonstrated that SD models can serve as effective feature extractors for various perception tasks.

**Diffusion Model.** Large-scale pre-trained diffusion models [15, 36, 37, 38, 16] are capable of producing high-quality images with reasonable structures and compositional semantics, indicating their robust spatial awareness and semantic understanding capabilities. Consequently, numerous studies have utilized pre-trained SD models as feature extractors, extracting features from images for visual perception tasks [39, 40, 41, 42]. Inspired by these works, [17] and [18] have employed the pre-trained SD model [15], as a representation learner for visual correspondence. In addition, [14] investigated the distinct properties of SD features compared to the DINO [35] feature, and jointly leveraged them for improved performance. Based on this, GeoAware-SC [43] proposes adaptive pose alignment to improve the geometric awareness of the SD features.

**Massive Activations.** Massive activations have been extensively studied in large language models (LLMs) [20, 44, 45]. Several works [20, 46] have shown that these activations are concentrated in a small number of fixed dimensions, particularly in the start and delimiter tokens. [47] further revealed that the massive values in the query and key vectors originate from the positional encoding mechanism RoPE[48]. In addition to LLMs, similar phenomena have also been observed in Vision Transformers (ViTs)[13, 49, 20]. Specifically, [20] identified attention artifacts in ViT feature maps, predominantly appearing in low-information background regions. DVT [49] further traced these artifacts to the influence of positional embeddings. Recent studies on accelerating DiTs [50, 51] have identified massive activation outliers in DiTs and revealed that these outliers lead to unstable model quantization and excessively high distillation losses during knowledge distillation.

## 3 Preliminaries

**Adaptive Layer Norm.** AdaLN operation $\text{AdaLN}(z; \gamma, \beta)$ adaptively normalizes feature $z$ with the adaptive scale $\gamma$ and shift $\beta$ parameters.

$$\hat{z} = (1 + \gamma) \, \text{LayerNorm}(z) + \beta \tag{1}$$

where $\text{LayerNorm}$ is the layernorm operation [52].

**DiT Architecture.** We follow the architecture used in [36]. DiT [36] comprises two parts: a VAE [53], which comprises an encoder $\mathcal{E}$ and a decoder $\mathcal{D}$ to project images from pixel space to latent space, and diffusion transformer blocks $\mathcal{A} = \{\mathcal{A}_k\}_{k=1}^N$, where k is the index and N is the number of blocks. Given an input image $x_0$, we first encode it to the latent representation $z_0 = \mathcal{E}(x_0)$ with the encoder $\mathcal{E}$ and then add corresponding noise $\epsilon$ to the latent to obtain a noisy latent representation $z_t \in \mathbb{R}^{C \times H \times W}$ according to a pre-defined timestep $t$. The DiT block $\mathcal{A}_k$ encodes the original feature $z_t^k$ as follows:

$$z_t^{k+1} = z_t^k + \alpha_k \mathcal{A}_k(z_t^k) \tag{2}$$

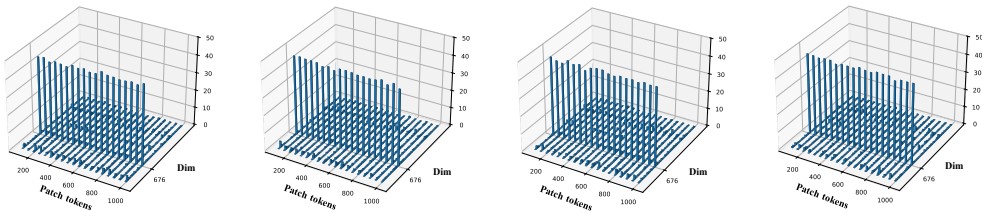

Figure 3: **Massive Activations in SD3-5.** We visualize the activation magnitudes of four different image features extracted using SD3-5. Notably, massive activations consistently distribute in a fixed dimension (676) across all image patch tokens.

where each Diffusion Transformer block employs residual connections [54], with $\alpha_k$ representing the residual scaling factor computed by the AdaLN-zero layer. It is important to note that the original feature throughout this paper is denoted as $z_t^k$, and our analysis of massive activations is conducted based on this representation.

**DiT block $\mathcal{A}_k$.** DiT block [36] includes a modulation layer called AdaIN-zero layers [36], which regress two groups of scale and shift parameters based on the given timestep $t$ and additional conditions (e.g., text embedding $c$).

$$\{(\gamma_k^i, \beta_k^i, \alpha_k^i)\}_{i=1}^2 = \text{MLP}_k(t, c) \tag{3}$$

where MLP is utilized to regress channel-wise scale and shift parameters $\gamma_k^i, \beta_k^i, \alpha_k^i \in \mathbb{R}^C$. $\gamma_k^i, \beta_k^i$ served as parameters for $i_{th}$ AdaLN and $\alpha_k^i$ is applied to scale $i_{th}$ residual connections.

Next, the block-input feature $z_t^k$ is passed through the Self-Attention layer to yield intermediate feature $z_t^{(k,2)}$.

$$\hat{z}_t^{(k,1)} = \text{AdaLN}(z_t^{(k,1)}; \gamma_k^1, \beta_k^1) \tag{4}$$

$$z_t^{(k,2)} = \text{Self-Attention}(\hat{z}_t^{(k,1)}) + \alpha_k^1 z_t^{(k,1)} \tag{5}$$

Finally, the intermediate feature $z_t^{(k,2)}$ is fed into the Feedforward layer to yield block-output feature $z_t^{(k+1,1)}$.

$$\hat{z}_t^{(k,2)} = \text{AdaLN}(z_t^{(k,2)}; \gamma_k^2, \beta_k^2) \tag{6}$$

$$z_t^{(k+1,1)} = \text{Feedforward}(\hat{z}_t^{(k,2)}) + \alpha_k^2 z_t^{(k,2)} \tag{7}$$

where $z_t^{(k,1)}$ and $z_t^{(k+1,1)}$ denote the same features as $z_t^k$ and $z_t^{k+1}$, respectively. For clarity, we refer to $z_t^{(k,1)}$ and $z_t^{(k,2)}$ as the pre-AdaLN intermediate features, and $\hat{z}_t^{(k,1)}$ and $\hat{z}_t^{(k,2)}$ as the post-AdaLN intermediate features in the subsequent sections. The residual scaling factor $\alpha_k$ in Equation (2) corresponds to $\alpha_k^2$. For simplicity, we use AdaLN to refer to the AdaLN-zero module in DiTs throughout this paper. See the Appendix A for an illustration of DiTs architecture [36].

# 4 DiTF: Diffusion Transformer Feature

In this section, we explore how to eliminate the undesirable effects of massive activations and extract semantically meaningful features from DiTs. Specifically, we first analyze the spatial and dimensional characteristics of massive activations in DiTs, then discuss why the AdaLN layer is well-suited to addressing massive activation in the context of representation learning, and finally present our approach for extracting semantic-discriminative features from DiTs.

## 4.1 Massive Activations in Diffusion Transformers

**Massive activations are high-magnitude scalar values.** We first introduce a quantitative criterion to characterize massive activations in DiT features. Following the definition proposed for LLMs [20], we adopt a generalized formulation: an activation is considered a massive activation if its magnitude is approximately 100 times greater than the median magnitude within its feature map. It is notable that massive activations are scalar values, which are determined jointly by the patch and dimensions [20].

Table 1: **Massive activations exhibit relatively low variance compared to non-massive activations.** We present the mean and variance of activation values across image patches for selected feature dimensions ranked by average magnitude. The 1st-ranked dimension (massive activations) shows substantially lower variance relative to its mean compared to the 10th and 20th (no-massive activations), indicating limited local information.

| Model | 1st | 2th | 10th | 20th | Median |
|-------|-----|-----|------|------|--------|
| SD3-5 | -44.51±0.50 | 2.03±1.39 | -0.18±2.36 | -0.25±2.07 | 0.21 |
| Flux | 40.66±3.99 | -31.62±8.50 | -0.41±3.16 | 0.49±2.92 | 0.13 |

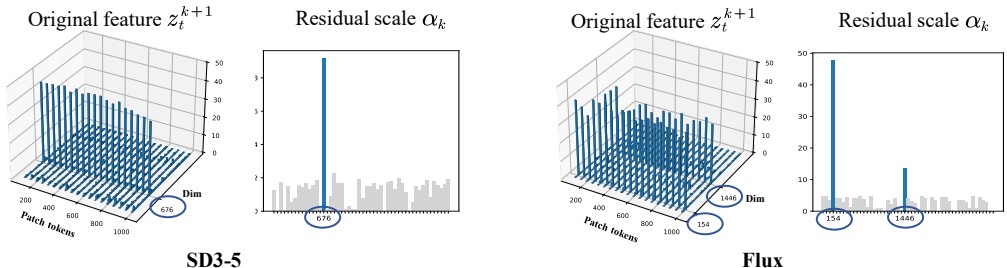

Figure 4: **Massive activations dimensions align with the residual scaling factor $\alpha_k$.** We visualize the magnitudes for the original feature $z_t^{k+1}$ and residual scaling factor $\alpha_k$.

**Massive activations are present in very few fixed dimensions.** We identify the dimensional locations of massive activations within the original DiT feature. As shown in Figure 3, the SD3-5 model exhibits massive activations concentrated in a single feature dimension (676). Similarly, in the Flux model, massive activations appear in two specific dimensions (154 and 1446). Analyses of other DiTs are provided in the Appendix. These observations suggest that massive activations consistently occur in a small number of fixed feature dimensions, which aligns with similar findings in LLMs [20].

**Massive activations appear across all spatial tokens.** As shown in Figures 2 and 3, massive activations consistently appear across all image patch tokens, regardless of the input image. For instance, in the SD3-5 model, the 676th feature dimension exhibits massive activations across all patch tokens. This phenomenon differs from the observations in LLMs and Vision Transformers (ViTs). Specifically, [20] reported that massive activations in LLMs are primarily concentrated in special tokens such as start and delimiter tokens, while [13] found that outlier activations in ViTs tend to emerge in low-information tokens.

**Massive activations hold little local information.** To better understand the nature of massive activations, we analyze their spatial variance by measuring the mean and variance of feature values across image patches. Feature dimensions are ranked by their averaged activation magnitude across patches. We then compare the top-ranked dimension (massive activations) with the 10th and 20th highest-ranked (non-massive) dimensions to assess their spatial discriminability. As shown in Table 1, the variance of the massive activations (1st) is substantially lower relative to its mean compared to the non-massive activations (10th and 20th), indicating that massive activations are less spatially discriminative. This suggests that massive activations encode limited local information.

**AdaLN enables accurate localization of massive activations.** It can be seen that the original features in DiTs are computed as $z_t^{k+1} = z_t^k + \alpha_k \mathcal{A}_k(z_t^k)$ via residual connections, where a scaling factor $\alpha_k$ is applied to weight the block output $\mathcal{A}_k(z_t^k)$ (see Equation (2)). To examine the potential link between massive activations and the residual scaling factor $\alpha_k$, we analyze their dimensional correspondence in Figure 4 and observe that they consistently co-occur in the same channels. Notably, $\alpha_k$ is regressed by the AdaLN-zero layer, suggesting that AdaLN-zero enables accurate localization of massive activations.

## 4.2 Channel-wise Modulation with AdaLN

Based on these observations, we hypothesize that AdaLN aggregates the activations in the block output into the dimension corresponding to the highest values in the scaling factor $\alpha_k$, resulting in dimension-concentrated massive activations. While such concentration may not be inherently

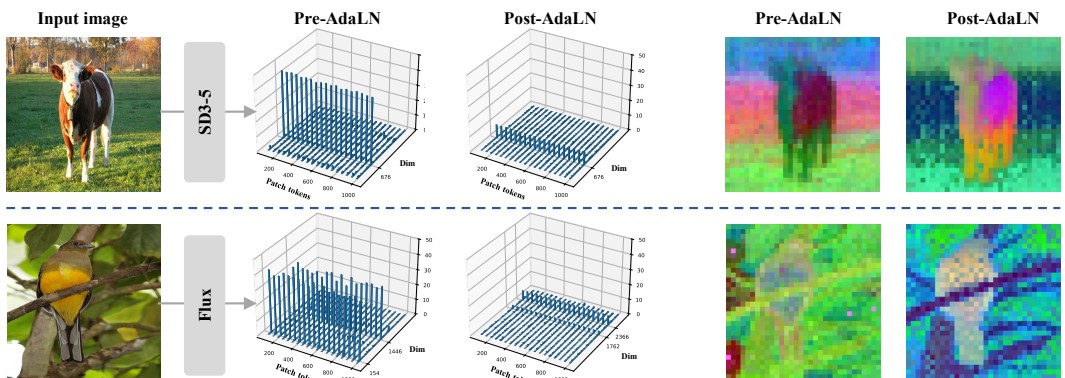

Figure 5: **AdaLN reduces massive activations and enhances feature semantics and discrimination.** We visualize the activation magnitudes and feature maps of both pre-AdaLN and post-AdaLN features in DiT block.

problematic, we argue that it is suboptimal for representation learning. Therefore, we further investigate how AdaLN interacts with the original features and massive activations within them.

**AdaLN alleviates massive activations.**  To better understand how AdaLN interacts with massive activations, we compare the properties of pre-AdaLN and post-AdaLN intermediate features within DiT blocks. As shown in Figure 5, massive activations in feature channels are significantly reduced after AdaLN processing. This suggests that AdaLN can effectively identify such concentrated activations and adaptively normalize them through channel-wise modulation, using the scaling and shifting parameters.

**AdaLN enhances feature spatial semantics and discrimination.**  Furthermore, we visualize different feature maps in Figure 5. It can be observed that the pre-AdaLN feature maps in DiTs exhibit limited semantic coherence, primarily capturing low-level textures such as object color (e.g., the cow and bird), and show weak discrimination between objects and background. In contrast, the post-AdaLN features display markedly improved spatial coherence and clearer semantic boundaries across object parts, suggesting that AdaLN improves spatial semantics and feature discrimination. More visualization results can be found in the Appendix, where the same phenomenon is observed.

## 4.3   Extracting features from Diffusion Transformers

Based on these observations, we propose to extract DiT features with the channel-wise modulation by AdaLN. Specifically, the feature extraction process in DiTs can be decomposed into two stages: (1) extracting the original feature $z_t^k$ from the DiT block $\mathcal{A}_k$, and (2) modulating it via adaptive channel-wise scaling and shifting using the AdaLN layer.

$$z_t^k = \mathcal{A}_k(z_t^{k-1}), \quad \gamma_k, \beta_k = \mathrm{MLP}_k(t, c) \tag{8}$$

$$\hat{z}_t^k = (1 + \gamma_k)\,\mathrm{LayerNorm}\left(z_t^k\right) + \beta_k \tag{9}$$

where $z_t^k$ is the intermediate pre-AdaLN feature, omitting the index $i$ for clarify. $\hat{z}_t^k$ denotes the final extracted feature. For simplicity, we refer to it as $\hat{z}$ in the following section.

**Channel discard.**  As shown in Figure 5, it can be observed that the post-AdaLN features still exhibit a few weakly massive activations, which will slightly compromise the local semantics of feature maps. Therefore, we propose a simple yet effective channel discard strategy to further suppress their influence. Specifically, we zero out the corresponding dimensions of these weakly massive activations in the post-AdaLN features as they hold little local information.

# 5   Experiments

## 5.1   Experimental Setup

**Model Variants.** We evaluate our strategies on four different pre-trained DiTs: Pixart-alpha [21], SD3 [22], SD3-5 [22], and Flux [23]. The channel-wise modulation with AdaLN operation for DiTs is conditioned on the input timestep $t$ and text embedding $c$, except for Pixart-alpha [21], which

Table 2: **Performance comparison on dataset SPair-71k.** We present per-class and average PCK@0.10 on the test split. The compared methods are divided into two categories: supervised (S) and unsupervised (U). We report PCK `per point` results for the (U) methods and PCK `per image` results for the (S) methods, following [43]. †: Integrating DINOv2 features for correspondence. The highest mAPs are highlighted in bold, while the second highest mAPs are underlined.

| | Method | Aer | Bike | Bird | Boat | Bottle | Bus | Car | Cat | Chair | Cow | Dog | Horse | Motor | Person | Plant | Sheep | Train | TV | V All |
|---|---|---|---|---|---|---|---|---|---|---|---|---|---|---|---|---|---|---|---|---|
| S | SCOT [55] | 34.9 | 20.7 | 63.8 | 21.1 | 43.5 | 27.3 | 21.3 | 63.1 | 20.0 | 42.9 | 42.5 | 31.1 | 29.8 | 35.0 | 27.7 | 24.4 | 48.4 | 40.8 | 35.6 |
| | PMNC [27] | 54.1 | 35.9 | 74.9 | 36.5 | 42.1 | 48.8 | 40.0 | 72.6 | 21.1 | 67 | 58.1 | 50.5 | 40.1 | 54.1 | 43.3 | 35.7 | 74.5 | 59.9 | 50.4 |
| | SCorrSAN [28] | 57.1 | 40.3 | 78.3 | 38.1 | 51.8 | 57.8 | 47.1 | 67.9 | 25.2 | 71.3 | 63.9 | 49.3 | 45.3 | 49.8 | 48.8 | 40.3 | 77.7 | 69.7 | 55.3 |
| | CATs++ [29] | 60.6 | 46.9 | 82.5 | 41.6 | 56.8 | 64.9 | 50.4 | 72.8 | 29.2 | 75.8 | 65.4 | 62.5 | 50.9 | 56.1 | 54.8 | 48.2 | 80.9 | 74.9 | 59.8 |
| | DHF [34] | 74.0 | 61.0 | 87.2 | 40.7 | 47.8 | 70.0 | 74.4 | 80.9 | 38.5 | 76.1 | 60.9 | 66.8 | 66.6 | 70.3 | 58.0 | 54.3 | 87.4 | 60.3 | 64.9 |
| | SD+DINO (S) [14] | 81.2 | 66.9 | 91.6 | 61.4 | 57.4 | 85.3 | 83.1 | 90.8 | 54.5 | 88.5 | 75.1 | 80.2 | 71.9 | 77.9 | 60.7 | 68.9 | 92.4 | 65.8 | 74.6 |
| U | ASIC [56] | 57.9 | 25.2 | 68.1 | 24.7 | 35.4 | 28.4 | 30.9 | 54.8 | 21.6 | 45.0 | 47.2 | 39.9 | 26.2 | 48.8 | 14.5 | 24.5 | 49.0 | 24.6 | 36.9 |
| | DINOv2+NN [12] | 72.7 | 62.0 | 85.2 | 41.3 | 40.4 | 52.3 | 51.5 | 71.1 | 36.2 | 67.1 | 64.6 | 67.6 | 61.0 | 68.2 | 30.7 | 62.0 | 54.3 | 24.2 | 55.6 |
| | DIFT [18] | 63.5 | 54.5 | 80.8 | 34.5 | 46.2 | 52.7 | 48.3 | 77.7 | 39.0 | 76.0 | 54.9 | 61.3 | 53.3 | 46.0 | 57.8 | 57.1 | **71.1** | 63.4 | 57.7 |
| | DiTF$_{sd3-5}$(ours) | 66.5 | 56.8 | 86.3 | 40.1 | 51.3 | 58.6 | 58.8 | 81.3 | **47.8** | 79.6 | 60.2 | 68.5 | 65.7 | 73.5 | **64.4** | 67.8 | 69.7 | **66.5** | 64.6 |
| | DiTF$_{flux}$(ours) | **74.3** | **65.0** | **88.1** | **48.1** | **53.2** | **60.7** | **60.7** | **84.9** | 42.4 | **82.8** | **68.4** | **72.1** | **70.9** | **74.2** | 62.1 | **72.6** | 66.0 | 60.3 | **67.1** |
| | SD+DINO† [14] | 73.0 | 64.1 | 86.4 | 40.7 | 52.9 | 55.0 | 53.8 | 78.6 | 45.5 | 77.3 | 64.7 | 69.7 | 63.3 | 69.2 | 58.4 | 67.6 | 66.2 | 53.5 | 64.0 |
| | GeoAware-SC† [43] | 78.0 | 66.4 | 90.2 | 44.5 | **60.1** | 66.6 | 60.8 | 82.7 | 53.2 | 82.3 | 69.5 | 75.1 | 66.1 | 71.7 | 58.9 | 71.6 | 83.8 | 55.5 | 69.6 |
| | DiTF$_{sd3-5}$(ours)† | **79.1** | 67.8 | **90.6** | 48.3 | 56.1 | **69.2** | **65.0** | **85.7** | **59.4** | **84.6** | **72.6** | **76.7** | 69.9 | 75.5 | 62.1 | **74.4** | **85.6** | **59.6** | **72.2** |
| | DiTF$_{flux}$(ours)† | 76.3 | 67.8 | 88.5 | **50.3** | 55.8 | 60.9 | 60.8 | 82.9 | 47.1 | 81.3 | 70.5 | 72.3 | **70.0** | **76.2** | **62.6** | 72.5 | 74.8 | 56.5 | 67.6 |

Table 3: **Performance comparison on datasets SPair-71k, AP-10K, and PF-Pascal datasets at different PCK levels.** We report the performance of the AP-10K intra-species (I.S.), cross-species (C.S.), and cross-family (C.F.) test sets. We report the PCK `per image` results following [43]. †: Integrating DINOv2 features for correspondence. The highest mAPs are highlighted in bold, while the second highest mAPs are underlined.

| | Method | SPair-71k | | | AP-10K-I.S. | | | AP-10K-C.S. | | | AP-10K-C.F. | | | PF-Pascal | | |
|---|---|---|---|---|---|---|---|---|---|---|---|---|---|---|---|---|
| | | 0.01 | 0.05 | 0.10 | 0.01 | 0.05 | 0.10 | 0.01 | 0.05 | 0.10 | 0.01 | 0.05 | 0.10 | 0.05 | 0.10 | 0.15 |
| S | SCorrSAN [28] | 3.6 | 36.3 | 55.3 | - | - | - | - | - | - | - | - | - | 81.5 | 93.3 | 96.6 |
| | CATs++ [29] | 4.3 | 40.7 | 59.8 | - | - | - | - | - | - | - | - | - | 84.9 | 93.8 | 96.8 |
| | DHF [34] | 8.7 | 50.2 | 64.9 | 8.0 | 45.8 | 62.7 | 6.8 | 42.4 | 60.0 | 5.0 | 32.7 | 47.8 | 78.0 | 90.4 | 94.1 |
| | SD+DINO (S) [14] | 9.6 | 57.7 | 74.6 | 9.9 | 57.0 | 77.0 | 8.8 | 53.9 | 74.0 | 6.9 | 46.2 | 65.8 | 80.9 | 93.6 | 96.9 |
| U | DINOv2+NN [12] | 6.3 | 38.4 | 53.9 | 6.4 | 41.0 | 60.9 | 5.3 | 37.0 | 57.3 | 4.4 | 29.4 | 47.4 | 63.0 | 79.2 | 85.1 |
| | DIFT [18] | 7.2 | 39.7 | 52.9 | 6.2 | 34.8 | 50.3 | 5.1 | 30.8 | 46.0 | 3.7 | 22.4 | 35.0 | 66.0 | 81.1 | 87.2 |
| | DiTF$_{sd3-5}$(ours) | 7.9 | 48.1 | 61.2 | 7.6 | **50.2** | **63.3** | 6.6 | 48.0 | 60.2 | **5.8** | 33.5 | 47.9 | 88.5 | 95.2 | 97.3 |
| | DiTF$_{flux}$(ours) | **8.3** | **49.3** | **64.0** | 7.6 | 49.7 | 62.2 | **6.8** | **48.2** | **61.7** | 5.7 | **34.2** | **48.9** | **89.5** | **95.8** | **97.6** |
| | SD+DINO† [14] | 7.9 | 44.7 | 59.9 | 7.6 | 43.5 | 62.9 | 6.4 | 39.7 | 59.3 | 5.2 | 30.8 | 48.3 | 71.5 | 85.8 | 90.6 |
| | GeoAware-SC† [43] | 9.9 | 49.1 | 65.4 | 11.3 | 49.8 | 68.7 | 9.3 | 44.9 | 64.6 | 7.4 | 34.9 | 52.7 | 74.0 | 86.2 | 90.7 |
| | DiTF$_{sd3-5}$(ours)† | **11.8** | **53.1** | **67.4** | **15.7** | **54.4** | **71.3** | **13.1** | **52.0** | **69.4** | **9.8** | **41.4** | **56.7** | **89.7** | **96.2** | **97.8** |
| | DiTF$_{flux}$(ours)† | 10.1 | 50.0 | 66.0 | 12.9 | 52.0 | 68.8 | 11.2 | 49.7 | 67.1 | 9.0 | 39.9 | 55.6 | 89.6 | 95.9 | 97.6 |

follows its AdaLN design and is conditioned only on timestep $t$. Additional experimental details, including DiT configurations and hyperparameter settings, are provided in Appendix C.

**Datasets and Evaluation Metric.** We evaluate our DiTF on semantic correspondence, geometric correspondence, and temporal correspondence. For semantic correspondence, we conduct experiments on three popular benchmarks: SPair-71k [57], PF-Pascal [58], AP-10K benchmark [43]. The AP-10K benchmark [43] is a new large-scale semantic correspondence benchmark built on AP-10K [59]. It comprises 2.61 million/36,000 data pairs, and spans three settings: the main intra-species set, the cross-species set, and the cross-family set. Following [18, 14], we adopt the percentage of correct key-points (PCK) metric. See Appendix for the results of two additional correspondence tasks.

## 5.2 Main results on semantic correspondence

To evaluate our model DiTF, we conducted two types of semantic correspondence experiments: in-category semantic correspondence on SPair-71k [57] and PF-Pascal [58], and cross-category semantic correspondence on AP-10K [43].

**In-category Semantic Correspondence** We conducted comprehensive experiments across different datasets where the results can be found in Tables 2 and 3. From these results, we could obtain the following observations: **1) State-of-the-art performance.** Overall, our models achieve state-of-the-art performance across different datasets. Specifically, our model DiTF$_{sd3-5}$(ours)† achieves 2.6% ↑ on Spair-71k and 2.6% ↑ on AP-10K-I.S. **2) Superior feature extraction capabilities.** It is readily apparent that our feature extraction models for DiTs, such as DiTF$_{sd3-5}$(ours) and DiTF$_{flux}$(ours), outperform traditional feature extractors like SD2-1 (DIFT [18]) and DINOv2 [12] across various datasets, demonstrating their strong and robust feature extraction abilities. Specifically, our DiTF$_{flux}$(ours) achieves a performance of 67.1% (9.4% ↑ over DIFT) on the Spair-71k dataset, and obtains a performance of 62.2% (1.3% ↑ over DINOv2) on dataset AP-10K-I.S. **3) Feature**

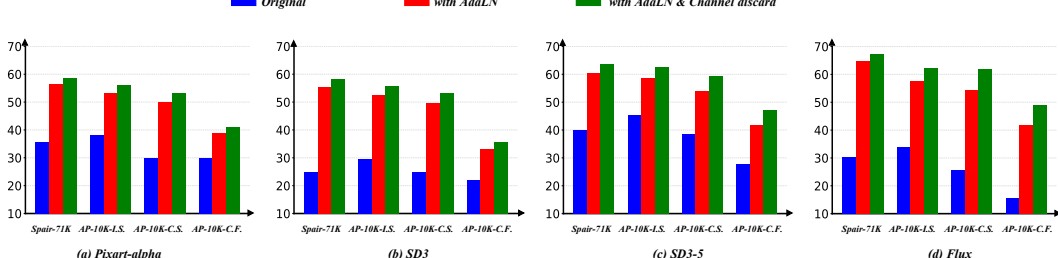

Figure 6: **Investigation of the impact of channel modulation with AdaLN and channel discard strategy across different datasets.**

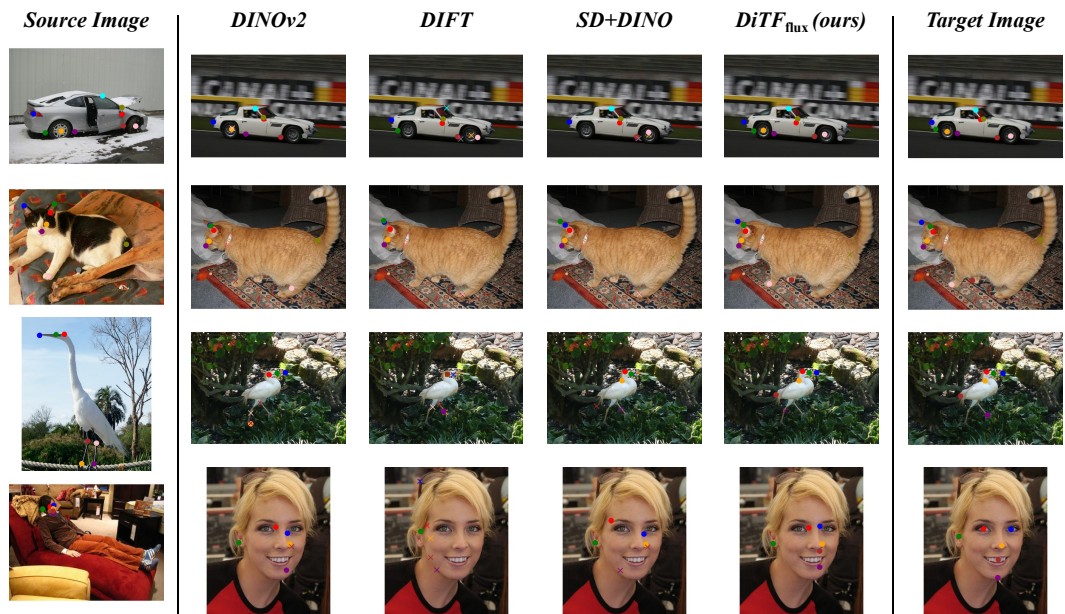

Figure 7: **Visualization of semantic correspondence prediction on SPair-71k using different features.** Different colors represent different key points where circles denote correctly predicted points under the threshold $\alpha_{bbox} = 0.1$ and crosses denote incorrect matches.

**redundancy across DiTs and DINOv2**: As shown in Table 2, when integrating DINOv2, our DiTF$_{\mathtt{sd3-5}}$ achieves a notable performance increase (from 64.6% to 72.2%). In comparison, our DiTF$_{\mathtt{flux}}$ shows a limited improvement, only a 0.5% gain (from 67.1% to 67.6%). We interpret this performance discrepancy from feature alignment. FLUX aligns closely with DINOv2, causing redundancy, while SD3.5 benefits more from DINOv2's complementary distribution. This alignment may arise from two factors: Improved alignment between diffusion models and DINOv2 with larger models and longer training [60]; Overlap in training data.

**Cross-category Semantic Correspondence** In addition, we conducted cross-category semantic correspondence experiments on the AP-10K dataset, as shown in Table 3. The results indicate that our models enable the extraction of accurate semantic features for cross-category image pairs and the drawing of correct correspondences, outperforming the previous best method, GeoAware-SC [43], by nearly 4.8% (64.6% to 69.4%)on AP-10K-C.S. Furthermore, the results show that previous methods, such as DINOv2 and DIFT, experience significant performance degradation on the cross-category task. In contrast, our model demonstrates stronger robustness for the cross-category correspondence.

## 5.3 Further analysis

**Evaluation of Channel Modulation with AdaLN.** We conducted comprehensive semantic correspondence experiments across different datasets on various DiTs. The results, as depicted in Figure 6 and Table 4, reveal several key observations: **1) Effectiveness.** When integrated with the channel modulation by AdaLN, all the DiT models experience an absolute performance boost of more than 20%, demonstrating that AdaLN is capable of effectively mitigating massive activations and

Table 4: **Ablation study of our model DiTF$_{flux}$** on the dataset SPair-71k and AP-10K. We report the PCK `per point` for SPair-71k and PCK `per image` for AP-10K-I.S.

| Model Variants | SPair-71k | | | AP-10K-I.S. | | |
|---|---|---|---|---|---|---|
| | 0.01 | 0.05 | 0.10 | 0.01 | 0.05 | 0.10 |
| Original | 2.3 | 14.6 | 29.9 | 2.4 | 16.5 | 33.9 |
| + AdaLN | 7.8 | 52.3 | 65.3 | 6.4 | 46.8 | 60.3 |
| + Channel discard | **8.3** | **56.3** | **67.1** | **7.6** | **49.7** | **62.2** |

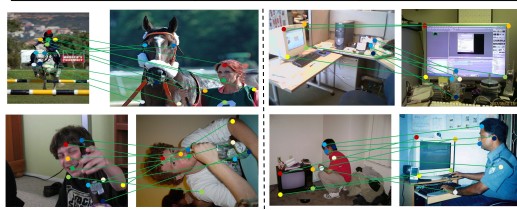

Figure 8: **Visualization of multi-to-multi correspondences with our model.** Green lines indicate correct matches and red incorrect

Table 5: **Conditions ablation study of AdaLN for feature channel-wise modulation** on SPair-71k. We report the PCK@0.10 `per point`.

| Condition | Pixart-Alpha | SD3 | SD3-5 | Flux |
|---|---|---|---|---|
| *original* | 35.6 | 24.8 | 39.8 | 29.9 |
| $c$ | - | 17.9 | 36.8 | 20.3 |
| $t$ | 55.3 | 54.6 | 58.2 | 60.9 |
| $t$ & $c$ | - | 54.9 | 58.3 | 63.6 |

Table 6: **Semantic segmentation on ADE20K.**

| Method | mIoU$^{ss}$ | mIoU$^{ms}$ |
|---|---|---|
| *self-supervised pre-training* | | |
| DINOv2 [12] | 47.7 | 53.1 |
| *SD-based pre-training* | | |
| VPD [42] | 53.7 | 54.4 |
| *DiTs-based pre-training* | | |
| DiTF$_{flux}$(w/o AdaLN) | 43.8 | 44.8 |
| DiTF$_{flux}$ | 53.6 | 54.8 |

enhancing feature semantics and discrimination. **2) Robustness.** These results indicate that channel modulation with AdaLN enables the accurate normalization of features across different categories, leading to significant improvements on cross-category datasets.

**Evaluation of Channel Discard strategy.** As shown in Figure 6 and Table 4. We can observe that our channel discard strategy can effectively locate and eliminate the weakly massive activations in the DiT feature across different DiTs, leading to smoother feature maps and higher correspondence matching accuracy.

**Impact of conditions in AdaLN for Channel-wise Modulation.** From the results in Table 5, we can observe that the timestep $t$ is essential for modulating massive activations, bringing a significant performance boost. In contrast, attempting to modulate features solely based on $c$ proves ineffective. This phenomenon arises from the inherent nature of DiTs. During training, DiTs introduce noise to images at varying levels of $t$, making the timestep an essential condition for feature modulation. It serves as a key factor in locating the channel positions of massive activations and enhancing feature quality, enabling their transformations into a clean and semantically meaningful representation.

**Qualitative Results.** We visualize some semantic correspondence predictions from different models in Figures 7 and 8. The visualization reveals that our DiT-based model yields robust and accurate correspondences across various complex scenes, including changes in viewpoint, multiple objects, and dense matching. These results demonstrate the superiority and robustness of our models.

## 5.4 Other Task: Semantic Segmentation

To assess the generalizability of DiTF, we conduct semantic segmentation experiments on ADE20K [61]. We extract activated DiT features using DiTF without applying the channel discard strategy and train a segmentation head following DINOv2 [12]. As shown in Table 6, DiTF$_{flux}$ outperforms DINOv2, with channel modulation of AdaLN significantly enhancing feature semantics and discriminability, leading to a substantial mIoU improvement (from 44.8 to 54.8). These results highlight the effectiveness and broad applicability of our approach.

## 6 Conclusion

In this paper, we identify and characterize massive activations in Diffusion Transformers (DiTs). We observe that these activations consistently emerge in a few fixed dimensions across all image patch tokens and carry limited local information. We further demonstrate that the built-in AdaLN mechanism in DiTs effectively suppresses massive activations while enhancing feature semantics and discriminability. Building on these insights, we propose a training-free AdaLN-based framework, Diffusion Transformer Feature (DiTF), which extracts semantically discriminative features from DiTs through channel-wise modulation using AdaLN. Extensive experiments on visual correspondence tasks validate the robustness and effectiveness of our approach.

# 7    Limitations and Future Work.

In this work, we primarily focus on understanding and mitigating massive activations in Diffusion Transformers (DiTs) from the perspective of representation learning. However, the emergence and potential roles of massive activations in the generative process of DiTs remain underexplored. We believe that investigating these activations from a generative viewpoint could provide deeper insights into their underlying mechanisms and potentially contribute to improving generative performance.

## Acknowledgements

The paper is supported in part by the National Natural Science Foundation of China (No. U21B2013, 62325109), and in part by the Shanghai 'The Belt and Road' Young Scholar Exchange Grant (24510742000). Mehrtash Harandi is supported by the Australian Research Council (ARC) Discovery Program DP250100262.

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

# Appendix

## A Diffusion Transformer Architecture

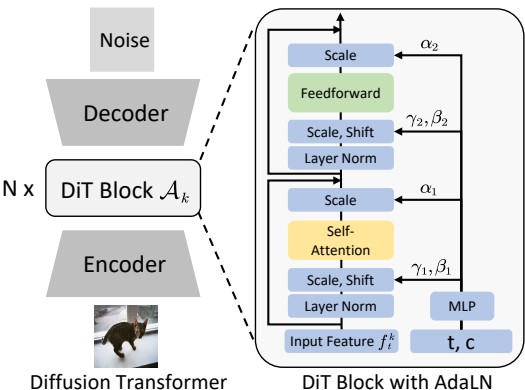

Figure 9: **DiT architecture illustration.**

We strictly follow the architecture used in DiT [36]. We provide an illustration of the DiT architecture as shown in Figure 9.

## B More Visualization Results

### B.1 Massive Activations in DiTs

To further illustrate the emergence of massive activations in DiTs, we provide additional visualization results in Figure 13. We show the LayerNorm-normalized activation magnitudes of original features from SD2-1 and various DiTs. While SD2-1 exhibits smooth activations, DiTs consistently show spikes concentrated in a few fixed dimensions across all patch tokens, revealing a fundamental difference that contributes to their degraded performance.

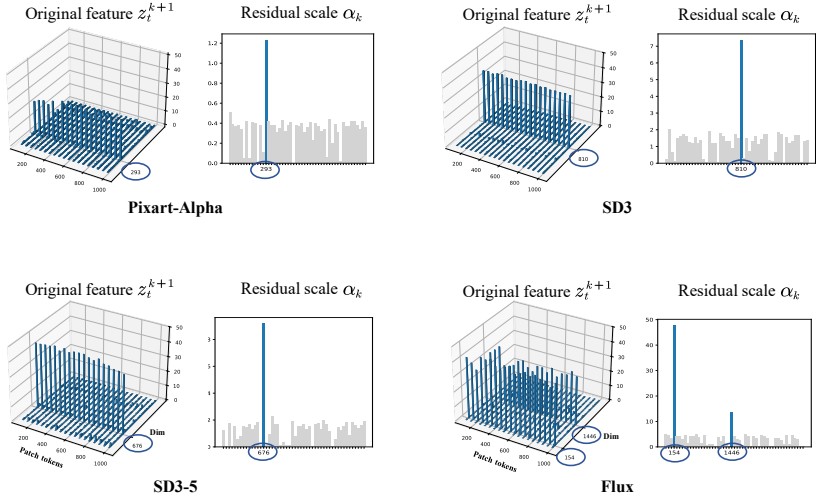

Figure 10: **Massive activations dimensions align with the residual scaling factor** $\alpha_k$. We visualize the magnitudes for the original feature $z_k^{t+1}$ and residual scaling factor $\alpha_k$.

### B.2 Massive Activations Dimensions Align with Residual Scaling Factor

In this section, we provide additional visualizations to examine the dimensional alignment between massive activations and the residual scaling factor $\alpha_k$ from the AdaLN layer. As illustrated in Fig-

ure 10, massive activations consistently co-occur with large values of $\alpha_k$ in the same dimensions across all DiTs.

## B.3   Channel-wise Modulation with AdaLN

To comprehensively demonstrate the impact of the built-in AdaLN in DiTs, we provide additional visualizations comparing pre-AdaLN and post-AdaLN features across various models, including SD3-5 (Figure 11), Pixart-Alpha (Figure 14), SD3 (Figure 15), and Flux (Figure 16). These results consistently show that AdaLN accurately localizes and normalizes massive activations, while enhancing feature semantics and discrimination through effective channel-wise modulation.

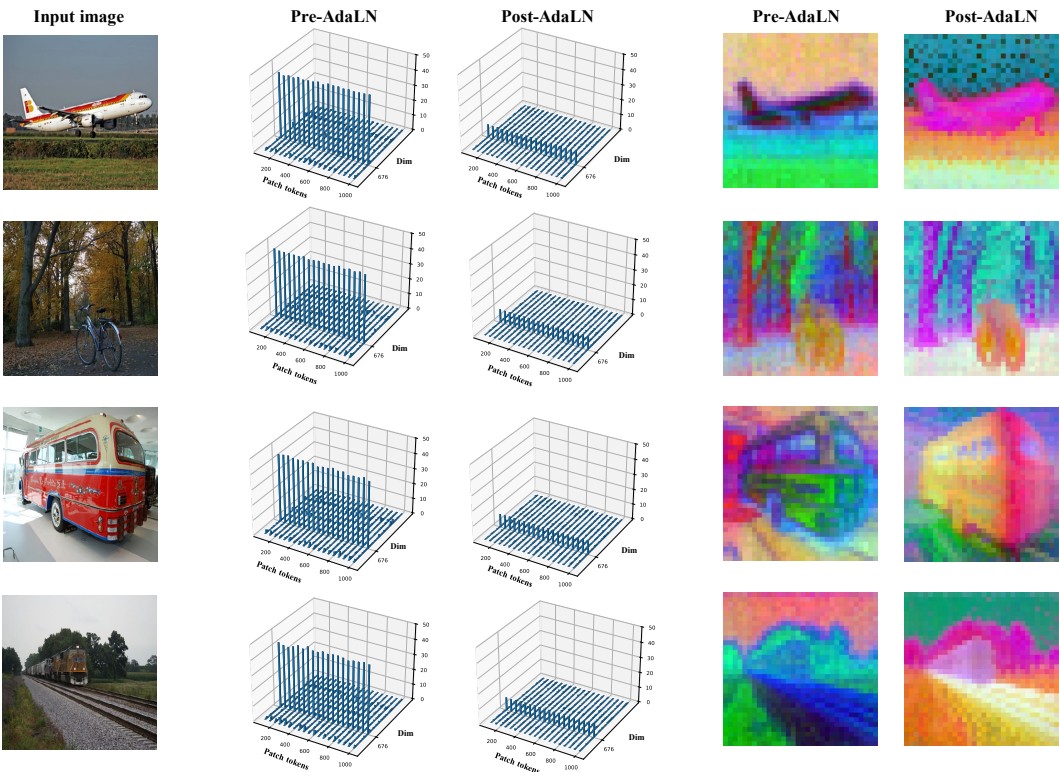

Figure 11: **Comparisons of pre-AdaLN and post-AdaLN features in SD3-5.**

## C   Further Implementation Details

**Configurations of different DiTs.** We employ the pre-trained Diffusion Transformers (DiTs) as a feature extractor for semantic correspondence. Formally, we decompose the universal feature extraction process in DiTs into two stages: (1) extracting the original feature $z_t^k$ from the DiT block $\mathcal{A}_k$, and (2) modulating it via adaptive channel-wise scaling and shifting using the AdaLN layer. For Pixart-alpha [21], SD3 [22], and SD3-5 [22], we first extract the pre-AdaLN feature $z_t^{(k,2)}$ and then activate it as follows.

$$z_t^{(k,2)} = \mathcal{A}_k(z_t^k), \quad \gamma_k^2, \beta_k^2 = \mathrm{MLP}_k(t, c) \tag{10}$$

$$\hat{z}_t^k = \left(1 + \gamma_k^2\right) \mathrm{LayerNorm}\left(z_t^{(k,2)}\right) + \beta_k^2 \tag{11}$$

As some of the Flux [23] model's blocks contain only one group of AdaLN-zero layer, we extract pre-AdaLN feature $z_t^{(k,1)}$ and then activate it as follows.

$$z_t^{(k,1)} = \mathcal{A}_k(z_t^k), \quad \gamma_k^1, \beta_k^1 = \mathrm{MLP}_k(t, c) \tag{12}$$

$$\hat{z}_t^k = \left(1 + \gamma_k^1\right) \mathrm{LayerNorm}\left(z_t^{(k,1)}\right) + \beta_k^1 \tag{13}$$

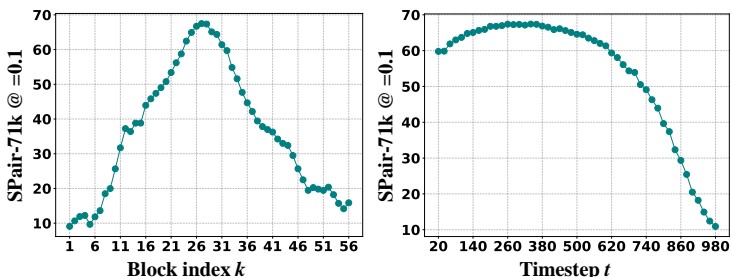

Figure 12: **Investigation of the DiT block index $k$ and timestep $t$.** We report the results of DiTF$_{\texttt{flux}}$ on dataset Spair-71k.

Table 7: **Configurations of different DiTs for semantic correspondence.**

| Method | Layers N | Hidden size d | Timestep $t$ | Block index $k$ |
|---|---|---|---|---|
| DiTF$_{\texttt{pixart-}\alpha}$ | 28 | 1152 | 141 | 14 |
| DiTF$_{\texttt{SD3}}$ | 24 | 1536 | 340 | 9 |
| DiTF$_{\texttt{SD3-5}}$ | 38 | 2432 | 380 | 23 |
| DiTF$_{\texttt{flux}}$ | 57 | 3072 | 260 | 28 |

The configurations of different DiTs can be found in Table 7, where the total time step T is 1000. We set the input image size as 960x960 for DiTs and 840x840 for the model DINOv2.

**Investigation on block index $k$ and timestep $t$.** The layer index $k$ and the timestep $t$ are critical hyperparameters that influence the quality of the extracted features from Diffusion Transformers (DiTs). Previous studies [18, 14] have conducted thorough investigations to identify the optimal layer index $k$ and timestep $t$ for Stable Diffusion. To explore the effects of varying $k$ and $t$ in DiTs, we conducted grid search experiments to identify the optimal hyperparameters. The results are presented in Figure 12. The figure reveals that features extracted from the middle layer achieve optimal performance across different DiTs. Furthermore, feature extraction in DiTs is robust to the timestep for semantic correspondence tasks, as a wide range of $t$ achieves excellent performance.

**Integration of DINOv2 feature.** We extracted DINOv2 features from the token facet of the 11th layer of the model. We then concatenated the DiT's features with the DINOv2 features in the channel dimension. To improve the efficiency of correspondence calculation, we computed Principal Component Analysis (PCA) across the pair of images for the features extracted from DiT, as follows:

$$\tilde{z}^s, \tilde{z}^t = \text{PCA}\left(\hat{z}^s \| \hat{z}^t\right) \tag{14}$$

where $\hat{z}^s, \hat{z}^t$ are the extracted source and target image DiT features. We only apply the PCA operation to SD3-5 and Flux features due to their high dimension and set the output dimension size as 1280.

## D  Geometric Correspondence

To comprehensively evaluate our model DiTF, we conduct additional experiments on geometric correspondence.

**Datasets.** Following [18], we evaluate our model on the HPatches benchmark [78], which comprises 116 sequences: 57 with illumination changes and 59 with viewpoint variations. Adopting the approach from CAPS [68], we detect up to 1,000 key points per image and apply cv2.findHomography() to estimate homography using mutual nearest neighbor matches.

**Metric.** We adopt the corner correctness metric for evaluation where we compute the average error between the four estimated corners of one image and the ground-truth corners with a threshold $\epsilon$ pixels, following [68, 18],.

**Results.** To comprehensively evaluate our models, we conducted geometric correspondence experiments on the HPatches benchmark [78], as detailed in Table 8. From the results, it can be observed that our model enables robust feature extraction for image pairs and draws precise geometric correspondence. Specifically, our model DiTF$_{\texttt{flux}}$(ours) achieve comparable performance 41.9% compared to the state-of-the-art model DIFT (SD-based). These results show that Diffusion Transformers can be employed as an effective feature extractor for geometric correspondence.

Table 8: **Geometric correspondence results on dataset HPatches.** We report the homography estimation accuracy [%] at 1, 3, 5 pixels.

| Method | Geometric Supervision | All | | | Viewpoint Change | | | Illumination Change | | |
|---|---|---|---|---|---|---|---|---|---|---|
| | | $\epsilon=1$ | $\epsilon=3$ | $\epsilon=5$ | $\epsilon=1$ | $\epsilon=3$ | $\epsilon=5$ | $\epsilon=1$ | $\epsilon=3$ | $\epsilon=5$ |
| SIFT [26] | None | 40.2 | 68.0 | 79.3 | 26.8 | 55.4 | 72.1 | 54.6 | 81.5 | 86.9 |
| LF-Net [62] | | 34.4 | 62.2 | 73.7 | 16.8 | 43.9 | 60.7 | 53.5 | 81.9 | 87.7 |
| SuperPoint [63] | | 36.4 | 72.7 | 82.6 | 22.1 | 56.1 | 68.2 | 51.9 | 90.8 | 98.1 |
| D2-Net [64] | Strong | 16.7 | 61.0 | 75.9 | 3.7 | 38.0 | 56.6 | 30.2 | 84.9 | 95.8 |
| DISK [65] | | 40.2 | 70.6 | 81.5 | 23.2 | 51.4 | 67.9 | 58.5 | 91.2 | 96.2 |
| ContextDesc [66] | | 40.9 | 73.0 | 82.2 | 29.6 | 60.7 | 72.5 | 53.1 | 86.2 | 92.7 |
| R2D2 [67] | | 40.0 | 74.4 | 84.3 | 26.4 | 60.4 | 73.9 | 54.6 | 89.6 | 95.4 |
| *w/ SuperPoint kp.* | | | | | | | | | | |
| CAPS [68] | Weak | 44.8 | 76.3 | 85.2 | 35.7 | 62.9 | 74.3 | 54.6 | 90.8 | 96.9 |
| DINO [12] | | 38.9 | 70.0 | 81.7 | 21.4 | 50.7 | 67.1 | 57.7 | 90.8 | 97.3 |
| OpenCLIP [69] | None | 33.3 | 67.2 | 78.0 | 18.6 | 45.0 | 59.6 | 49.2 | 91.2 | 97.7 |
| DIFT [18] | | **45.6** | **73.9** | **83.1** | **30.4** | **56.8** | **69.3** | 61.9 | **92.3** | **98.1** |
| DiTF$_{\texttt{flux}}$(ours) | | 41.9 | 70.7 | 79.5 | 22.0 | 50.8 | 63.4 | **62.5** | 91.3 | 96.2 |

Table 9: **Tempral correspondence results on DAVIS-2017.** We report the region-based similarity $\mathcal{J}$ and contour-based accuracy $\mathcal{F}$ for DAVIS. Pre-: Pre-trained on videos.

| Pre- | Method | Dataset | DAVIS | | |
|---|---|---|---|---|---|
| | | | $\mathcal{J}\&\mathcal{F}_{\mathrm{m}}$ | $\mathcal{J}_{\mathrm{m}}$ | $\mathcal{F}_{\mathrm{m}}$ |
| ✓ | MAST | YT-VOS [70] | 65.5 | 63.3 | 67.6 |
| | SFC [71] | | 71.2 | 68.3 | 74.0 |
| | InstDis [70] | | 66.4 | 63.9 | 68.9 |
| | MoCo [73] | | 65.9 | 63.4 | 68.4 |
| | SimCLR [74] | ImageNet [72] | 66.9 | 64.4 | 69.4 |
| | BYOL [75] | w/o labels | 66.5 | 64.0 | 69.0 |
| ✗ | SimSiam [76] | | 67.2 | 64.8 | 68.8 |
| | DINO [32] | | 71.4 | 67.9 | 74.9 |
| | OpenCLIP [69] | LAION [77] | 62.5 | 60.6 | 64.4 |
| | DIFT [18] | | 70.0 | 67.4 | 72.5 |
| | DiTF$_{\texttt{flux}}$(ours) | | **72.2** | **69.2** | **75.1** |

# E    Temporal Correspondence

In addition, we conduct experiments to verify the temporal correspondence capability of our DiTF. Specifically, we investigate DiTF's performance on video object segmentation and pose tracking tasks, employing DiTs as a feature extractor for correspondence.

**Datasets.** We conduct experiments on the challenge video dataset: DAVIS-2017 video instance segmentation benchmark [79], following [18].

**Metric.** Following [18, 80], we adopt the region-based similarity $\mathcal{J}$ and contour-based accuracy $\mathcal{F}$ as the performance metric where we segment the nearest neighbors between the consecutive video frames based on the representation similarity.

**Results.** The temporal correspondence results can be found in Table 9. From the results, it can be observed that our model exhibits a superior capability to extract video frame features for temporal correspondence. Specifically, our model achieves 72.2% on the dataset DAVIS, which outperforms the previous state-of-the-art DIFT by 2.2%, demonstrating the superior effectiveness of our model.

# F    Qualitative Results on AP-10K

We show the qualitative comparison of our Diffusion Transformer model DiTF with both Stable Diffusion (SD) and SD+DINO [14] in AP-10K intra-species (Figure 17), cross-species ( Figure 18), and cross-family ( Figure 19) subset.

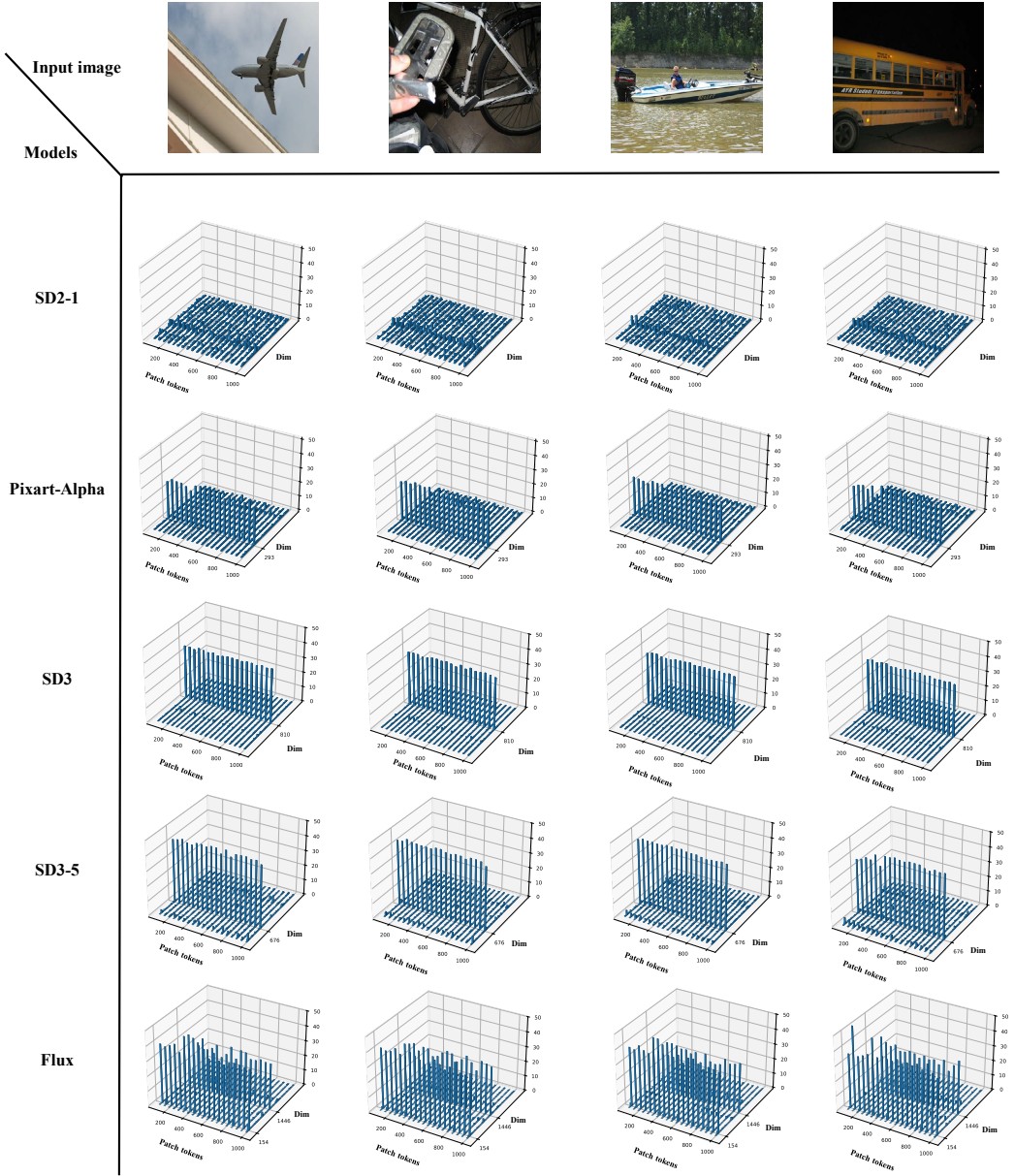

Figure 13: **Massive activations in DiTs.** SDv2-1 does not suffer from massive activations. However, all DiTs exhibit massive activations, which concentrate on very few fixed dimensions across all image patch tokens.

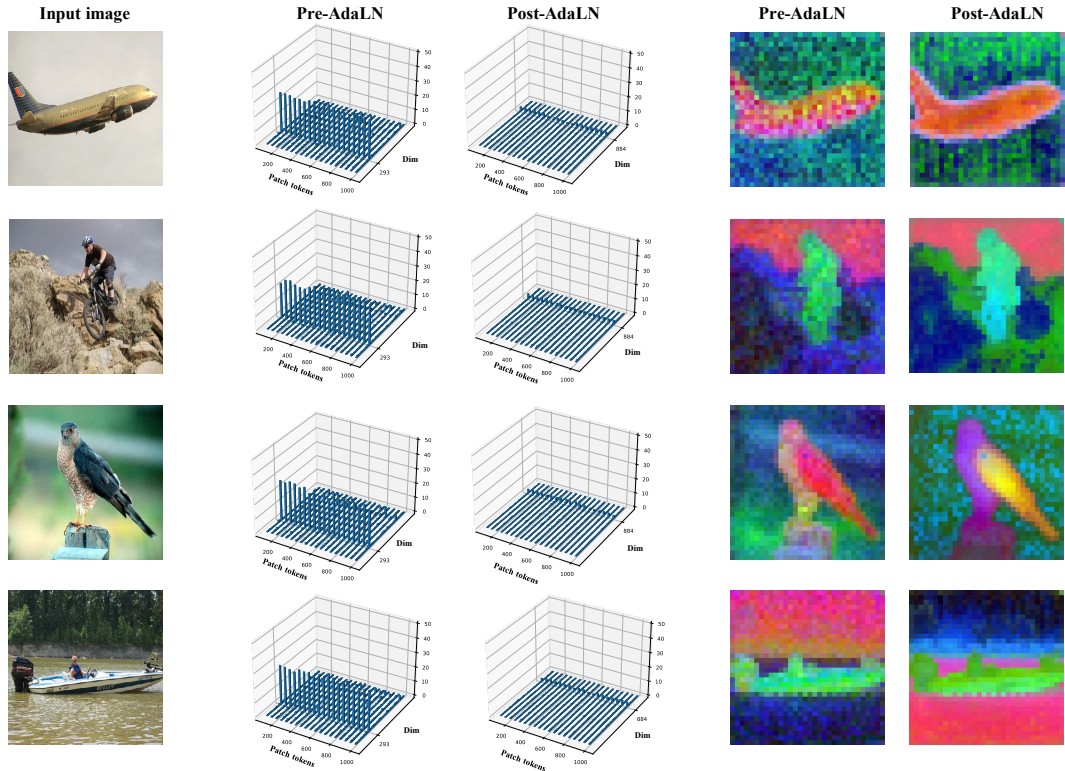

Figure 14: **Comparisons of pre-AdaLN and post-AdaLN features in Pixart-Alpha.**

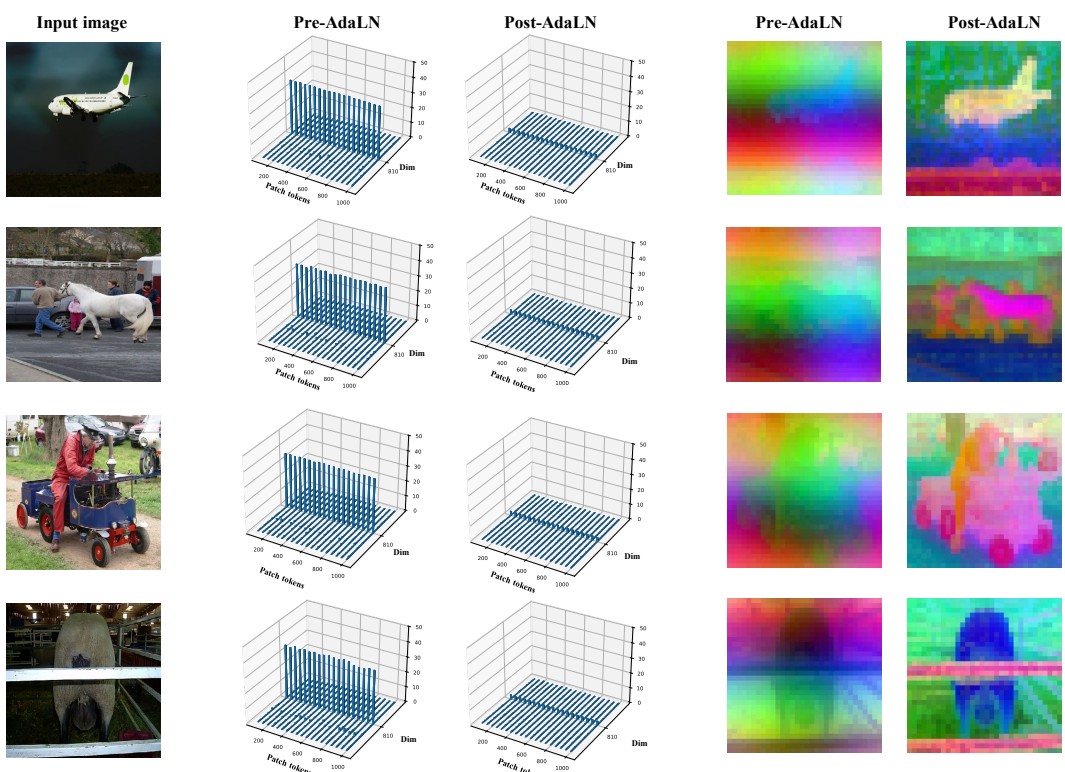

Figure 15: **Comparisons of pre-AdaLN and post-AdaLN features in SD3.**

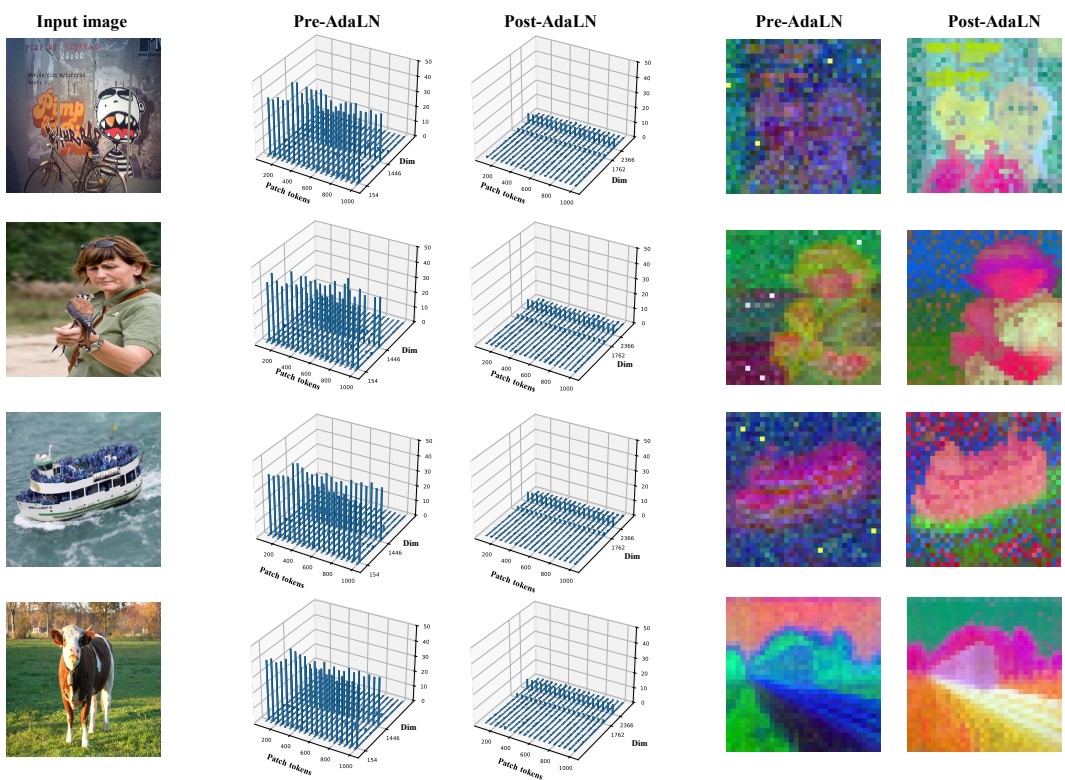

Figure 16: **Comparisons of pre-AdaLN and post-AdaLN features in Flux.**

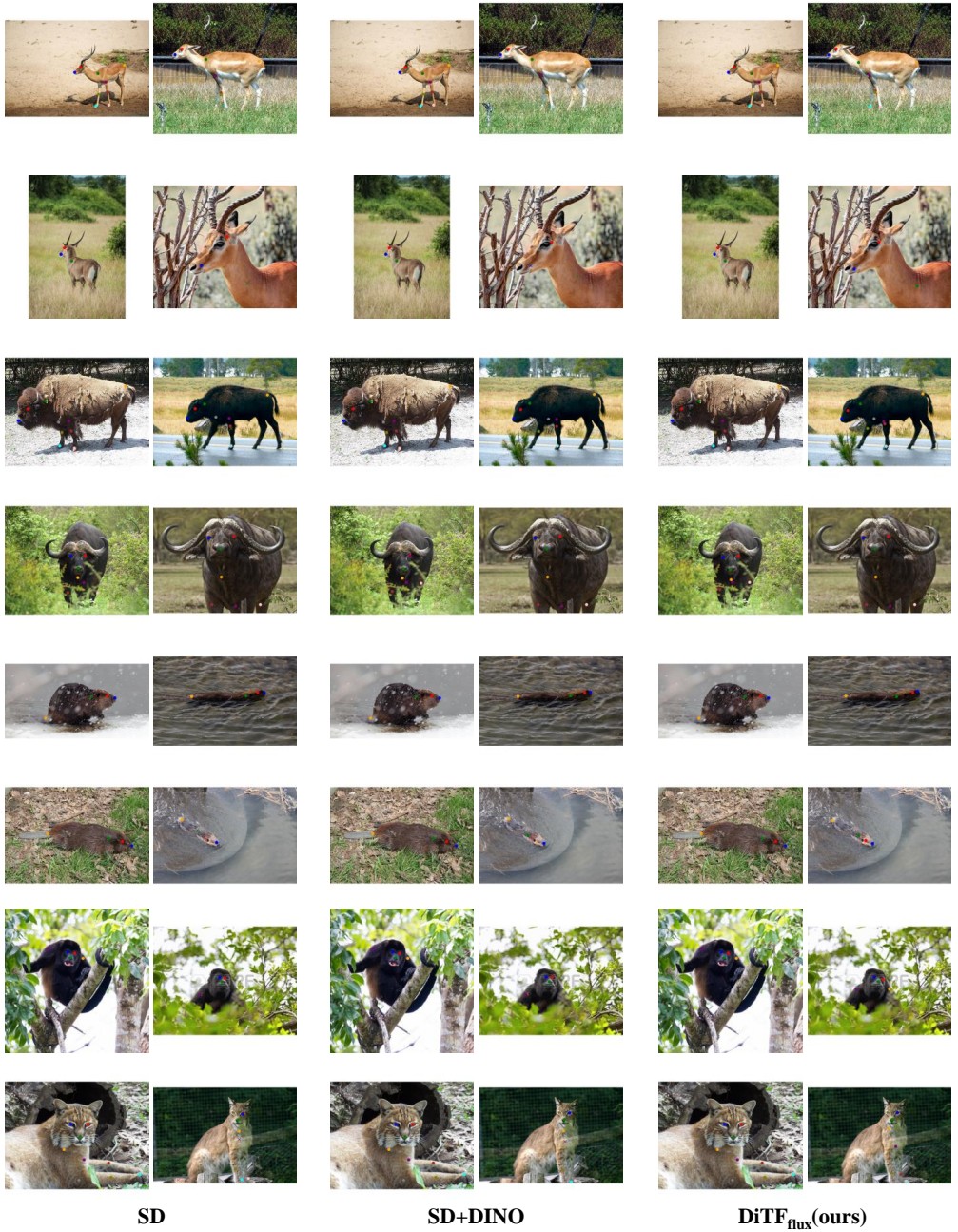

| SD | SD+DINO | DiTF$_{flux}$(ours) |

Figure 17: **Qualitative comparison on the AP-10K intra-species set.** Different colors represent different key points where circles denote correctly predicted points and crosses denote for incorrect matches.

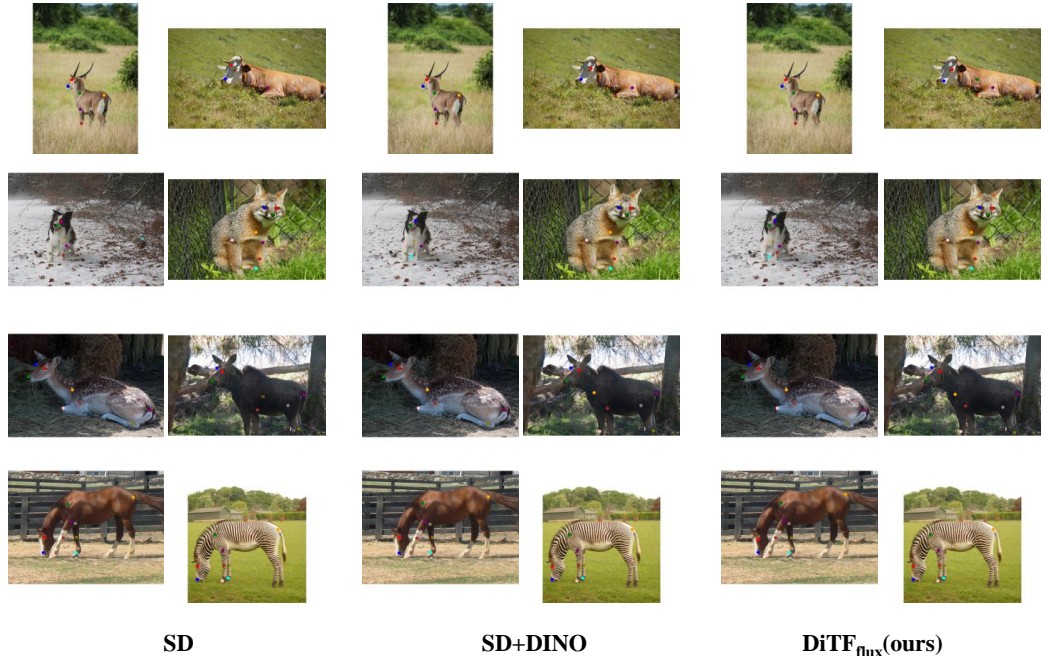

SD        SD+DINO      DiTF$_{\text{flux}}$(ours)

Figure 18: **Qualitative comparison on the AP-10K cross-species set.**

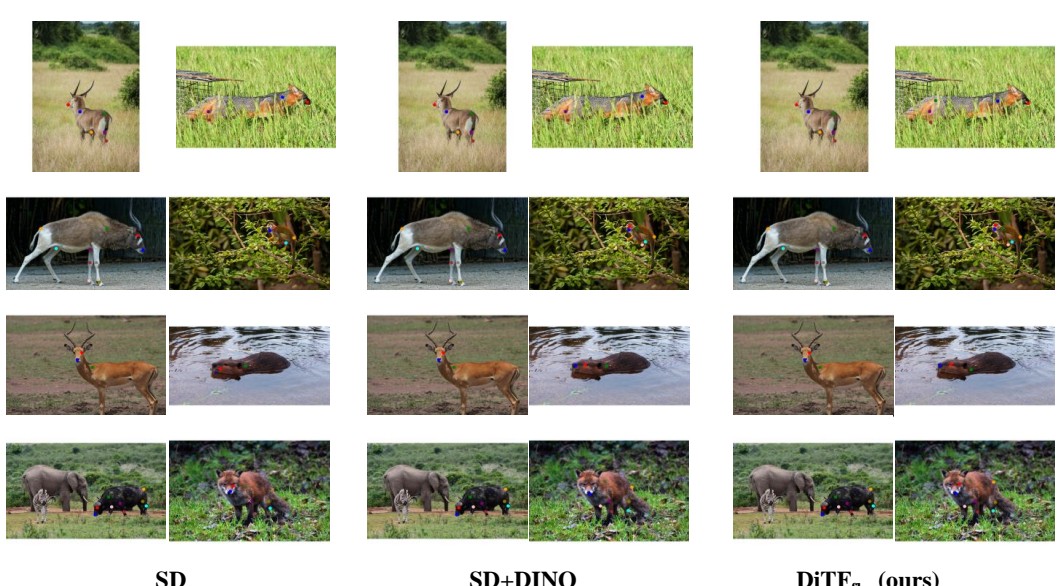

SD        SD+DINO      DiTF$_{\text{flux}}$(ours)

Figure 19: **Qualitative comparison on the AP-10K cross-family set.**

