# OpenReview forum: "Unleashing Diffusion Transformers for Visual Correspondence by Modulating Massive Activations"
_NeurIPS.cc/2025/Conference — NeurIPS 2025 poster_

### Official Review · Reviewer_gPKV · 2025-07-02

**Clarity:** 4
**Significance:** 3
**Originality:** 4
**Rating:** 5
**Confidence:** 4

**Summary:**

This paper investigates Diffusion Transformers (DiT) and demonstrates that a slight test-time modification significantly boosts their performance on correspondence tasks. Firstly, the paper reveals the presence of massive activation in DiT features and shows that AdaLN alleviates this issue, leading to improved feature representation. Furthermore, the paper employs channel discarding to enhance these representations. The authors tested their method on various standard correspondence benchmarks and also show performance on two variants of DiT, namely SD3-5 and Flux.

**Questions:**

This is a very interesting paper with a strong presentation, and I support its acceptance. However, the reason why massive activation is harmful for representation remains unclear. I have suggested a hypothesis, which may or may not be correct, and would appreciate the authors' comments and a test of this hypothesis. Whether my hypothesis is right or wrong, if the authors can provide another persuasive hypothesis, I would be happy to raise my score.

**Ethical Concerns:**

["NO or VERY MINOR ethics concerns only"]

**Final Justification:**

I greatly appreciate the authors' response. All of my concerns have been resolved. This is a good paper with excellent presentation and a highly interesting finding.

**Limitations:**

yes

**Paper Formatting Concerns:**

No formatting concerns.

**Quality:**

3

**Strengths And Weaknesses:**

**Strengths**

I have always wondered why DiT features exhibit weak performance on correspondence tasks, and this paper provides a compelling explanation through intriguing and well-supported observations, presented in a clearly written text. The finding that massive activation in DiT harms its matching ability is particularly interesting.

**Major Weaknesses**

The paper offers weak intuition regarding why features with massive activation are suboptimal for representation learning, as claimed in Line 166. The authors should at least elaborate on why this might be the case. Additionally, this argument could be considered an overstatement, given that diffusion features are typically well-learned from vast amounts of data. In my opinion, massive activation is specifically more detrimental to correspondence tasks. Typically, feature matching involves normalization followed by a dot product to construct cost volumes. My hypothesis is that since massive activation forms in the exact same channel for all features, the dot product between any features would be dominated by the square of this massive activation, leading to imprecise matching.

Based on the hypothesis above, I wonder how the performance would be if channel discarding were applied before AdaLN. If my hypothesis is correct, then the crucial point is not whether pre-AdaLN or post-AdaLN is used, but rather that channel discarding is the most significant solution. If this shows similar performance to DiTF, then the methodology section should be slightly modified to de-emphasize the effect of AdaLN and instead highlight the effectiveness of channel discarding itself.

**Minor Weaknesses**

- In Line 73, the paper that used GAN should also be cited.
- In Figure 5, the upper-left side of the activation visualization is partially cropped.
- It would be great if the temporal correspondence results were highlighted in the main text, not just in the supplementary material.
- Adding an explanation for why the geometric correspondence performance is lagging behind DIFT would be greatly appreciated. I suspect DIFT's high-resolution features might contribute to its strength in geometric correspondence.

---

> ### Author Rebuttal · Authors · 2025-07-30
>
> Thanks for your valuable suggestions. We’re glad that you found our work insightful and interesting. Please see the responses to your comments.
>
> ### **Q1. Reason why massive activation is suboptimal for representation**
> Thank you for your valuable comments. First, we would like to clarify that we have provided insightful intuition in the Introduction section (Lines 35–39), e.g., “These massive activations share a few fixed dominant dimensions, resulting in a high degree of **directional similarity** for feature vectors.” We are pleased to see that you agree with our explanation. To further investigate the effect of massive activations, we conduct detailed experiments on Table i, and we elaborate on several key findings based on the results.
>
> **1) Massive activations significantly hinder spatial discrimination**
>
> Due to the **directional homogeneity** introduced by massive activations, cosine similarity becomes ineffective in distinguishing spatial feature vectors. As shown in Table i, discarding the massive activation dimensions in feature maps significantly improves correspondence performance (from 29.9 to 37.5). Moreover, since DiTs are originally pretrained for generation tasks, their features often contain fine-grained visual details, which may be irrelevant or even detrimental for dense matching tasks [1]. For example, as illustrated in Figure 5 of the main text and Figures in the supplementary material, the original features (pre-AdaLN) tend to preserve vivid color cues as well as lighting and shading, which are **essential for generation but suboptimal** for visual correspondence (especially semantic correspondence). Consequently, the co-existence of massive activations and irrelevant details weakens the representational robustness of DiT features for correspondence tasks.
>
> **2) AdaLN enhances representation by suppressing activations and irrelevant details**
>
> As shown in Table i, directly applying channel discard to the original DiT features provides only suboptimal improvement, suggesting that reducing massive activations alone is insufficient. In contrast, AdaLN not only suppresses the massive activations but also effectively **activates other critical channels** essential for dense perception tasks. This dual effect mitigates the impact of massive activations while simultaneously **suppressing irrelevant low-level details** (e.g., color and shading), leading to a more refined and semantically meaningful feature space (see Figure 5 and the supplementary material). Consequently, the collaboration between AdaLN and channel discard operation leads to substantial gains in performance (**29.9 to 67.1**).
>
> | Model| Accuracy |
> | ----------------------------------- | :-------------------: |
> | Original|29.9|
> | with Channel discard |37.5|
> | with AdaLN|65.3|
> | with AdaLN & Channel discard|67.1|
>
> *Table i. Ablation study of our model DiTF$_{Flux}$ on the SPair-71k semantic correspondence benchmark.*
>
>
> ### **Q2. Explanation for the geometric correspondence performance**
>
> Thank you for your comments. We agree that feature resolution plays a critical role in determining geometric correspondence performance. Specifically, given an input image of size $H \times W$, our DiTF framework produces features at a resolution of $\frac{H}{16} \times \frac{W}{16}$. In contrast, DIFT [2] leverages higher-resolution features of size $\frac{H}{8} \times \frac{W}{8}$, which are extracted from its 2nd upsampling block. These higher-resolution features preserve finer-grained texture and structural details, thereby contributing to better geometric localization. As shown in Table ii, when DIFT utilizes the lower-resolution features of size $\frac{H}{16} \times \frac{W}{16}$ from its 1st upsampling block, it achieves a reduced accuracy of 37.2%, which is lower than our DiTF$_{Flux}$ (41.9%). This comparison underscores the effectiveness of our framework.
>
> | Model | Resolution| Accuracy |
> | :-------------: | :-------------------: | :-------------------: |
> | DIFT |$\frac{H}{8} \times \frac{W}{8}$|45.6|
> | DIFT|$\frac{H}{16} \times \frac{W}{16}$|37.2|
> | DiTF$_{Flux}$ (Our)|$\frac{H}{16} \times \frac{W}{16}$|41.9|
>
> *Table ii. Comparisons between DIFT and Our DiTF$_{Flux}$ model on the HPatches geometric correspondence benchmark.*
>
> ### **Q3. Typos and citations**
>
> Thank you for your feedback. In the final version, we will properly cite the related GAN literature and fix all typos and errors in Figure 5. In addition, we will highlight temporal correspondence results in the main text.
>
> [1] Representation alignment for generation: Training diffusion transformers is easier than you think, ICLR 2025.
>
> [2] Emergent Correspondence from Image Diffusion, NeurIPS 2023.

---

> > ### Comment · Reviewer_gPKV · 2025-08-05
> >
> > All of my concerns have been resolved. It would be great if the intuition behind why features with massive activation lead to suboptimal results were also highlighted in the methods section in more detail. I'll raise my score accordingly.
> >
> > I have one small suggestion regarding the method name: wouldn't "DIFT" and "DiTF" be confused with each other if presented in the same table due to the similarity of their names? It would be great if the method name were more distinct from "DIFT." This is entirely up to the authors' discretion.

---

> > > ### Author Response · Authors · 2025-08-05
> > >
> > > Thank you very much for your positive feedback and for acknowledging that your concerns have been addressed. We appreciate your suggestion to elaborate more on the intuition behind why features with massive activations lead to suboptimal results in the Methods section. We will take care to highlight this aspect more clearly in the final revision to improve clarity.
> > >
> > > Regarding the potential confusion between the method names “DiTF” and “DIFT,” we appreciate your thoughtful observation. We will carefully consider your suggestion and, if appropriate, adjust the naming to ensure clearer distinction in the presentation. Thank you again for your constructive comments.

---

### Official Review · Reviewer_92HZ · 2025-07-02

**Clarity:** 4
**Significance:** 3
**Originality:** 4
**Rating:** 5
**Confidence:** 3

**Summary:**

This paper investigates the usage of pre-trained Diffusion Transformers (DiTs) for visual correspondence tasks and identifies a phenomenon of "massive activations" in DiT feature maps. To mitigate this, the authors propose a training-free feature extraction method (DiTF) that leverages existing AdaLN layers and introduces a simple channel discard strategy to suppress such activations. The paper demonstrates consistent improvements across several benchmarks.

**Questions:**

If the authors can address my main concerns in the rebuttal, I will adjust my rating accordingly:

1. Perhaps I misunderstood, but could the authors clarify the distinction (if any) between the proposed AdaLN and the AdaLN that already exists in DiT models? As it stands, this appears to be reused rather than redesigned.

2. Your work suppresses or even zeros out massive activations. However, the Massive Activation paper you cite argues that these activations are critical for model performance. Why do you observe the opposite? Could you elaborate on this contradiction?

**Ethical Concerns:**

["NO or VERY MINOR ethics concerns only"]

**Final Justification:**

Please see my final response

**Limitations:**

yes

**Quality:**

4

**Strengths And Weaknesses:**

Strengths
The identification and quantification of massive activations in DiTs is the first attempt in the community. Ablation studies and visualizations are massive. The performances are evaluated across several datasets with consistent gains. The reference is sufficient and comprehensive, and up-to-date.

Weaknesses
1. The main concern is that the core contribution is relatively limited and heavily dependent on prior works, which undermines the novelty and significance required for a NeurIPS-level submission. It's more like a reimplementation of [20] in DiT [36] without novel insights. The entire pipeline builds upon existing DiT architectures and the concept of massive activations. The only novel contribution appears to be the channel discard strategy, which, while useful, cannot reach the bar for NeurIPS.

3. Misleading attribution of contribution: Several operations, particularly the use of AdaLN for normalization and modulation, are not proposed by this paper, but are rather built-in functionalities of DiTs. While authors claim that 'DiTF employs AdaLN to adaptively localize and normalize massive activations with channel-wise modulation,' which might be misleading.

4. Lack of theoretical depth: Many conclusions are derived from visual inspection or statistical correlations without formal analysis or proof. There is no theoretical justification for why the proposed modifications improve semantic representation. E.g., Massive activations hold little local information; however, the claim is made based on the low variance of the massive activations on the feature, which cannot directly lead to the conclusion.

5. Inconsistency with prior literature: According to the prior work [20] “Massive Activations in LLMs”, massive activations are correlated with important representational capacity, disabling them will lead to performance degradation. However, this paper argues for suppressing them entirely (even zeroing them out) to improve the performances, which appears contradictory and is not justified. It's strongly recommended to give a further explanation or analysis.

6. Since the proposed method is a universal paradigm, not specified for the task, it would be beneficial to also test it on some common CV tasks, like image classification.


Minors: The manuscript would benefit from a thorough proofreading—e.g., in line 107, “AdaIN” appears to be a typo and should be corrected. [45] seems to be a wrong citation where Massive activations were not discussed. Incorrect assumptions: The paper simplifies AdaLN-zero to AdaLN, ignoring the important differences in their initialization schemes. This is conceptually incorrect.

Final Justification
While the experimental results are decent, the paper relies too much on prior architectural elements and empirical heuristics. The channel discard strategy, being the only new technical contribution, is not substantial enough to warrant acceptance at NeurIPS. The work would benefit significantly from deeper theoretical insights or architectural innovations rather than retrofitting observations on existing models.

---

> ### Author Rebuttal · Authors · 2025-07-30
>
> Thank you for your thoughtful review. We appreciate your recognition of our contributions on massive activations in DiTs and address your concerns as follows:
>
> ### **Q1. Clarification of core contributions and novelty**
> We would like to clarify that our method is **fundamentally different** from a simple reimplementation of [1] within the DiTs, and it introduces novel insights and contributions beyond them. In particular, we emphasize the following key aspects:
>
> **1) Unique characteristics and insightful analysis of massive activations in DiTs.**
>
> - Compared to [1], we identify **distinctive characteristics** of massive activations in DiTs, which **differ fundamentally** from those observed in large language models (LLMs) [1]. In DiTs, the massive activations are consistently concentrated in a few fixed dimensions **across all spatial tokens** (see Figures 2 and 3). In contrast, massive activations in LLMs typically occur in **specific input tokens** (e.g., starting token).
> - Furthermore, our analysis reveals that the unique characteristic of massive activations in DiTs is closely associated with the **channel-wise residual scaling factors** introduced by AdaLN-Zero. Specifically, DiT feature updates are computed via residual connections: $z_{t}^{k+1} = z_{t}^{k} + \alpha_k A_k(z_{t}^{k}),$ where we observe a strong alignment between the dimensions exhibiting massive activations and those corresponding to large values of the residual scaling factors $\alpha_k$ (see Figure 4 and supplementary material). Such theoretical and empirical investigations demonstrate an **insightful connection** between massive activations and the scaling behavior of AdaLN-zero (recognized by Reviewer ntGm).
>
> **2) A reasonable and practical training-free framework for feature extraction with DiTs.**
>
> - Motivated by the observed relationship between massive activations and the residual scaling factors of AdaLN-Zero, we propose a **reasonable and elegant** feature extraction framework, DiTF, which can be directly applied to existing pre-trained DiTs **without requiring any additional training** (recognized by Reviewer ntGm). Specifically, we reuse the built-in AdaLN modules in DiTs to effectively identify and regulate these massive activations with adaptive channel modulation, significantly enhancing the quality of representations.
> - Furthermore, **strategies for handling massive activations in LLMs [1] do not transfer effectively to DiTs**. Specifically, when directly applying the activation intervention strategy in LLMs [1] to DiTs, it fails to improve feature quality, as evidenced in Table i (29.9 vs. 30.1), highlighting the **fundamentally different nature** of massive activations in LLMs and DiTs. In contrast, our proposed DiTF enhances both feature semantics and discriminability, leading to substantial performance improvements on dense spatial tasks (**29.9 to 67.1**).
>
> | Model| Accuracy |
> | ----------------------------------- | :-------------------: |
> | Flux |29.9|
> | Flux + Activation intervention in LLMs [1] |30.1|
> | Flux + DiTF (Our)|67.1|
>
> *Table i. Performance comparison of Flux models with different massive activation tackling strategies on the SPair-71k semantic correspondence benchmark*
>
> **3) A novel perspective on the poor performance of DiT features in dense tasks.**
> - When directly using DiT features for dense spatial tasks, it yields disappointing results (Figure 1). In this paper, we identify the massive activations within DiT feature maps as **a key factor** limiting performance (recognized by Reviewer gPKV). Unlike Stable Diffusion (SD2-1), DiTs exhibit massive activations concentrated in a few fixed dimensions **across all image patch tokens** (Figure 2). These dominant dimensions cause **high directional similarity** among feature vectors [3], which makes it difficult to distinguish them using cosine similarity, and ultimately degrades performance on dense spatial tasks (Lines 35-39).
>
> **4) Novel insights into future DiT training.**
> - **REPresentation Alignment (REPA)**: REPA [4] aligns DiT hidden states with pretrained features (e.g., DINOv2) to facilitate training. However, due to the distinct feature characteristics: DiTs exhibit massive activations whereas DINOv2 does not, direct alignment may be suboptimal. Instead, aligning features after AdaLN modulation may  improve compatibility and alignment quality.
>
>
> ### **Q2. On Theoretical depth and justification**
>
> We respectfully clarify that our conclusions are not solely based on visual inspection or simple correlations, but are derived from a rigorous and systematic study across various pre-trained DiT models and are supported by both comprehensive results and theoretical insights. Specifically:
>
> **1) Robustness and generality.**
>
> We examined a wide range of publicly available pre-trained DiT models, including SD3, SD3-5, Flux, etc., and consistently observed the same characteristics of massive activations. This consistent pattern suggests that our findings are not model-specific or incidental, but rather reflect a general property of DiTs.
>
> **2) Comprehensive experimental and visual support.**
>
> Our paper conducts systematic experiments complemented by carefully designed visualizations to thoroughly investigate the phenomenon of massive activations. Furthermore, we provide plentiful supporting evidence in the supplementary materials, collectively establishing a robust experimental foundation for our conclusions.
>
> **3) Theoretical justification for the proposed solution.**
>
> Our method is **theoretically motivated and interpretable**:
> - **Massive activation analysis**: We have provided theoretical justification for massive activation analysis in the introduction section (Lines 35-41). These massive activations share a few fixed dominant dimensions across all spatial tokens, resulting in a high degree of **directional similarity** for feature vectors. This property makes it challenging to distinguish spatial feature vectors using cosine similarity, which **hinders spatial discrimination of representation**. In addition, this analysis theoretically supports the conclusion that "Massive activations hold little local information."
> - **Connection to AdaLN-Zero**: DiT feature updates are computed via residual connections: $z_{t}^{k+1} = z_{t}^{k} + \alpha_k A_k(z_{t}^{k}),$ meaning AdaLN-zero directly affects the dominance and expressiveness of DiT features with residual scaling factor $\alpha_k$.
> - **Channel-wise modulation with AdaLN**: Motivated by the above insights, we employ the built-in AdaLN to modulate massive activations in DiT features, formulated as:
> $$
> \text{AdaLN}(z_{t}^{k}) = (1 + \gamma_k) \cdot \text{LayerNorm}(z_{t}^{k}) + \beta_k,
> $$
> where $\gamma_k \in \mathbb{R}^C$ and $\beta_k \in \mathbb{R}^C$ are adaptive channel-wise scaling and shifting parameters. Specifically, AdaLN effectively **identifies and suppresses** dimensions associated with massive activations through the scaling factor $(1 + \gamma_k)$, thus enhancing representation semantics and discrimination.
>
> ### **Q3. Clarifying the Alleged Contradiction with 'Massive Activations in LLMs'**
>
> We would like to clarify that our method **does not conflict with [1]**, as the **operational setting** and the **nature of massive activations** are fundamentally different.
>
> - In [1], the authors directly intervene in the hidden states during inference by manually setting massive activation values to zero in a single layer, and then **propagating the modified hidden state through the remaining model**. This intervention aims to probe the functional role of massive activations in the **internal computation** of LLMs. In contrast, our method treats the Diffusion Transformer (DiT) as a **frozen feature extractor**. We extract features from an intermediate block, apply AdaLN to these features, and perform a channel discard (zero out) operation to weakly massive activations in the **extracted features only**. Importantly, no modifications are made to the internal computation of the model.
> - Furthermore, the characteristics of massive activations in DiTs are **notably different** from those in LLMs [1]. The massive activations in LLMs [1] typically arise at **specific tokens** (e.g., starting token). In DiTs, such massive activations consistently occur at a **small number of fixed channel dimensions**, and appear **across all spatial tokens** (Figures 2 and 3). These **massive activations with concentration across all spatial tokens** in DiTs result in a high degree of **directional similarity** for feature vectors in the latent space [2]. Our proposed channel discarding strategy specifically addresses this issue by suppressing dominant but uninformative massive activations, thereby promoting feature discrimination and semantics.
>
>
> ### **Q4. Generalizability on various CV tasks**
>
> We have conducted **semantic segmentation** experiments on ADE20K (Table 6 of main text). Specifically, our approach boosts mIoU **from 43.8 to 53.6**. We further conduct **linear probing** on **ImageNet classification** using globally pooled DiT features, following the DINOv2 [3] protocol, and achieve a notable accuracy gain **from 65.8 to 72.6** (*DiTF$_{Flux}$*), highlighting the effectiveness and generalizability of our method.
>
> ### **Q5. Typos and citations**
> Thank you for your valuable feedback. In the final revision, we will correct all the typos and citation mistakes. In addition, we will revise our assumptions of AdaLN-Zero and AdaLN to accurately reflect their differences.
>
> [1] Massive activations in large language models, COLM 2024.
>
> [2] Clearclip: Decomposing clip representations for dense vision-language inference, ECCV 2024.
>
> [3] DINOv2: Learning Robust Visual Features without Supervision, TMLR 2025.
>
> [4] Representation Alignment for Generation: Training Diffusion Transformers Is Easier Than You Think, ICLR2025.

---

> ### Comment · Reviewer_92HZ · 2025-08-04
> **Very satisfied with the response from the authors**
>
> Thank you for the clarification. I find the authors’ explanation—along with the newly added experiments—both clear and convincing. The distinction between their method and [1], particularly in terms of operational settings and the nature of massive activations, is well articulated. It is especially helpful that the authors emphasize how their method operates on extracted features in a non-invasive manner (I refer to it more like a masking operation in this work instead of the Set to zero operation in [1]), without intervening in the model’s internal computations, leading to different performances (30.1 and 67.1).
>
> The additional experiment also strengthens the justification for their approach by highlighting the differences in activation patterns between LLMs and DiTs (i.e., token-specific vs. channel-specific and spatially consistent). The motivation for channel discarding is now well supported by the observed characteristics of massive activations in DiTs, and the intended impact on improving feature discrimination is clearly demonstrated.
>
> I strongly encourage the authors to explicitly incorporate this clarification and the new experimental evidence into the revised manuscript. Doing so will greatly help readers understand the key distinctions between their method and related work.
>
> Just one additional concern. While I understand that LLMs and vision models exhibit different massive activation patterns, I would appreciate it if the authors could further elaborate on why such differences also exist within vision models. Specifically, in [1], ViT and DINO do not exhibit massive activations concentrated in a few fixed dimensions across all spatial tokens, as observed in DiT in this work. Furthermore, the zero-set operation in [1] leads to a severe performance drop in ViT-based models, whereas in DiT, the same operation appears to yield a slight performance improvement.
>
> Overall, I appreciate the thoughtful and detailed rebuttal. As my main concerns are mostly addressed (difference to [1] and openration clarification), the method demonstrates both conceptual soundness and strong empirical performance with a high-quality, I will update my score to reflect this positive clarification based on the final response accordingly.

---

> > ### Author Response · Authors · 2025-08-05
> >
> > We sincerely appreciate your recognition of our work and are pleased to have addressed your main concerns. We will incorporate this clarification and the new experimental evidence into the final revised manuscript.
> >
> > Furthermore, we provide further elaborations on the differences of massive activations in LLMs, ViTs and DiTs.
> >
> > **Distinct training paradigms and supervisory signals result in different spatial location of massive activations in LLMs, ViTs and DiTs.**
> > - **Large Language Models (LLMs)**: LLMs are typically trained using an **autoregressive next-token prediction** objective, where each token prediction is conditioned on all preceding tokens. As the model generate the entire sequence token-by-token, the early tokens play a **disproportionately important role** in shaping the final output distribution. Consequently, the initial tokens (particularly starting token) tend to accumulate stronger activations, resulting in token-specific massive activations [1].
> > - **Vision Transformers (ViTs)**: For models such as CLIP [2] and DINOv2 [3], supervision is predominantly applied at the **global semantic level**, typically via the CLS token. For instance, CLIP uses contrastive learning between image and text embeddings based solely on the CLS token, while other patch tokens receive no direct supervision. Similarly, DINOv2 relies on self-supervised learning that aligns global semantics, again focusing supervision on the CLS token. This **uneven supervisory signal** leads ViTs to implicitly exploit redundant background tokens to store global semantic cues [4], resulting in localized massive activations concentrated in a subset of tokens rather than uniformly distributed across all image tokens.
> > - **Diffuson Transformers (DiTs)**: In contrast, DiTs are trained under a **denoising diffusion** paradigm, which enforces **spatially uniform supervision** across all image patches. Each patch token is treated equivalently, with the model learning to reconstruct all patches simultaneously based on the others. This uniform training objective leads to a **spatially uniform distribution** of massive activations across all image patch tokens.
> >
> > - To verify our hypothesis, we conduct a comparative analysis of massive activations in two large language models **LLaMA3-8B** [5] and **LLaDA-8B** [6]. Notably, these two models share a similar architecture, and the primary difference lies in their training paradigms: LLaMA3-8B follows **next-token prediction** objective, while LLaDA-8B is trained using **masked diffusion modeling**. As shown in Table i, massive activations in LLaMA3-8B are concentrated in a **specific token**: the starting token (<|startoftext|>). In contrast, LLaDA-8B exhibits massive activations in dimension 3848 **across all text tokens**. This distinction aligns with our theoretical claim that the choice of training paradigm fundamentally influences the activation distribution patterns.
> >
> >
> > | Text tokens |<\|startoftext\|>|"Summer"|$~~$"is" |$~~$"warm"|$~~$"."|"Winter"|$~~$"is"|$~~$"cold"|$~~$"."|
> > | :--: | :--: |:--: |:--: |:--: |:--: |:--: |:--: |:--: |:--: |
> > | LLaMA3-8B (788) | 226.0|2.0|1.4|0.9|1.2|0.2|1.4|0.7|2.3|
> > | LLaDA-8B (3848)|134.0|112.5|101.0|103.5|108.5|116.5|80.0|119.5|92.5|
> >
> > *Table i. Distribution of activation values along the massive activation dimension in LLaMA3-8B and LLaDA-8B. The dimension associated with massive activations is indicated by (·). The input sentence: "Summer is warm. Winter is cold.".*
> >
> >
> > **Zero-set operation in [1] leads to a severe performance drop in ViT-based models.**
> >
> > For ViT-based models (e.g., CLIP, DINOv2), massive activations typically emerge in specific tokens (redundant background tokens), which tend to encode **global semantic** information [4]. Consequently, applying a zero-set operation to suppress these activations can severely disrupt the global representation, resulting in a significant performance drop on tasks that rely on holistic understanding, such as image classification.
> >
> > In contrast, DiTs exhibit massive activations across all spatial tokens, leading to a high degree of **directional similarity** among feature vectors in the latent space. Suppressing these dominant yet uninformative activations enhances feature discrimination and semantic clarity, yielding performance improvements on spatial visual correspondence tasks.
> >
> >
> >
> >
> > [1] Massive activations in large language models, COLM 2024.
> >
> > [2] Learning Transferable Visual Models From Natural Language Supervision, PMLR 2021.
> >
> > [3] DINOv2: Learning Robust Visual Features without Supervision, TMLR 2025.
> >
> > [4] Vision Transformers Need Registers, ICLR 2024.
> >
> > [5] The llama 3 herd of models.
> >
> > [6] Large Language Diffusion Models.

---

> > > ### Author Response · Authors · 2025-08-05
> > >
> > > **ViTs Also Exhibit Distinct Massive Activation Behaviors under Different Supervision Strategies**
> > >
> > > To further illustrate the spatial locations of massive activations in ViTs, we present a more detailed analysis in Table ii. Specifically, we conduct a comparative study of two Vision Transformer models: DINOv2 [1] and MAE [2]. Notably, both models share the same backbone architecture (ViT-L), with the key difference lying in their training paradigms—DINOv2 adopts a self-supervised framework with **sparse, global semantic supervision**, whereas MAE is trained through **dense masked token reconstruction**.
> > >
> > > ****
> > >
> > > | Model |1st|2th |10th|20th|Median|
> > > | :--: | :--: |:--: |:--: |:--: |:--: |
> > > | DINOv2 (471) | 348.2|18.3|16.0|14.3|5.7|
> > > | MAE (927)|101.6|83.5|76.5|72.3|55.4|
> > >
> > > *Table ii. Distribution of activation values along the identified massive activation dimension in DINOv2 and MAE. The dimension exhibiting massive activation is denoted by (·). For each model, we report both the activation values and the corresponding median values across different image tokens within this massive activation dimension.*
> > >
> > > As shown in Table ii, massive activations in DINOv2 are concentrated in a **single token**, whereas in MAE, massive activations appear in feature dimension(927) **across all image tokens**. This observation highlights that **different supervision strategies can lead to distinct patterns of massive activation**, even when using the same ViT architecture. Specifically, when only a subset of tokens (e.g., the CLS token) is supervised as in DINOv2, the model tends to produce localized massive activations. In contrast, when all input tokens are supervised uniformly in a spatial manner as in MAE, the resulting massive activations are distributed across all tokens.
> > >
> > >
> > > [1] DINOv2: Learning Robust Visual Features without Supervision, TMLR 2025.
> > >
> > > [2] Masked Autoencoders Are Scalable Vision Learners, CVPR2022.

---

> > > > ### Comment · Reviewer_92HZ · 2025-08-05
> > > >
> > > > Thank you for the detailed and professional rebuttal. I truly appreciate the authors' thorough response and the additional experimental analysis provided in Table ii. The comparison between DINOv2 and MAE convincingly illustrates that different supervision strategies can indeed lead to distinct patterns of massive activation, even when the backbone architecture remains the same. This insight significantly deepens our understanding of massive activations in ViTs, especially in relation to token-level supervision.
> > > >
> > > > The additional experiments can enhance the technical depth of the work, which not only presents new findings but also opens up several new directions for future investigation (e.g., how training strategy affects the massive activation behaviors). The clarity of writing, the depth of analysis, and the novel perspectives explored in the work all reflect the authors’ deep research insight and commitment to advancing the field. I strongly recommend adding those insights into the main body and inspiring the following readers.
> > > >
> > > > Considering both the technical quality and the superb performance of the proposed method, I'm glad to adjust my rating and recommend accepting this paper.

---

> > > > > ### Author Response · Authors · 2025-08-06
> > > > >
> > > > > Thank you for your positive feedback and constructive suggestions. We are glad that the additional analysis has fully addressed your concerns regarding ViT-based models. We will follow your advice and incorporate these insights into the main paper to benefit future readers. We truly appreciate your support and recommendation.

---

### Official Review · Reviewer_f4nx · 2025-07-03

**Clarity:** 3
**Significance:** 3
**Originality:** 3
**Rating:** 5
**Confidence:** 4

**Summary:**

The authors identify and analyze  “massive activation” in Diffusion Transformers (DiT): certain feature's activation exhibit significantly large activations that infected fine-grained tasks such as visual correspondence. To address this, they propose DiTF (Diffusion Transformer Feature), a modified DiT framework incorporating AdaLN, an adaptive LayerNorm that learns channel scaling and shifting parameters conditioned on the activation statistics. Also they discard the weakly massive activation after the AdaLN features by zero out the corresponding dimensions. Experiments on standard benchmarks demonstrate that DiTF substantially reduces activation peaks and variance, leading to around 5–10 % higher matching accuracy in semantic correspondence. This work is the first to diagnose massive
activations in DiT, offering a general normalization design that enhances stability and performance in large scale diffusion Transformer models

**Questions:**

1. In Section 4.2, it is stated that AdaLN alleviates massive activations. However, in Figure 5, after applying DiTF to the Flux model, the dimensions of the original massive activations and the normalized (weakened) massive activations appear to be inconsistent. Could you explain why this occurs, and whether it is reasonable?
2. As mentioned in the weaknesses, could you provide additional comparisons of the SD3.5 and Flux  models on specific tasks before and after applying DiTF? I did not find these results in the paper’s Results section. Also, which version of the SD3.5 model are you using? (SD3.5 large or Medium)

**Ethical Concerns:**

["NO or VERY MINOR ethics concerns only"]

**Final Justification:**

The author addressed my concerns properly in their rebuttal. Based on the new discussion and explanation, I decided to raise my rating.

**Limitations:**

This work lacks an analysis of its methodological limitations; the authors could examine potential constraints of their approach, including its impact on transferability to a broader set of tasks.

**Paper Formatting Concerns:**

1. Figure 1a and Figure 2 can be color coded.

**Quality:**

3

**Strengths And Weaknesses:**

Strengths:
1. This paper carefully evaluates the “massive activation” phenomenon in the DiT model and provides a detailed analysis of it.
2. The exploration of the phenomenon and the application of the proposed method follow a logical flow, and the experimental results are relatively comprehensive.

Weakness:
1. The paper’s explanation of how AdaLN specifically mitigates the “massive activation” phenomenon is not sufficiently detailed. The evaluation metrics lack direct comparisons between the same base models (e.g., SD3.5 or Flux) with and without the DiTF framework

---

> ### Author Rebuttal · Authors · 2025-07-30
>
> Thanks for your suggestions and for recognizing the novelty and contribution of our work. Please see the responses to your comments.
>
> ### **Q1. Details of how AdaLN mitigates the massive activations**
>
> Thank you for your valuable comments. We have described the channel-wise modulation strategy using AdaLN to address massive activations in Sections 4.2 and 4.3. To further clarify the mitigation process, we elaborate on the rationale and mechanism below.
>
> **1) Why utilize AdaLN to mitigate the massive activations?**
>
> In DiTs, feature updates are computed via residual connections: $z_{t}^{k+1} = z_{t}^{k} + \alpha_k A_k(z_{t}^{k}),$ where $\alpha_k$ denotes the channel-wise residual scaling factor from the AdaLN-Zero layer. We observe that the massive activation dimensions in $z_{t}^{k+1}$ strongly align with large values of $\alpha_k$ in the corresponding channels (as shown in Figure 4). This reveals an **inherent connection** between massive activations and the scaling behavior of AdaLN. Based on this insight, we leverage the adaptive nature of AdaLN to **identify and regulate** these massive activations.
>
> **2) How to mitigate the massive activations with AdaLN?**
>
> The modulation process of AdaLN is formalized as:
> $$
> \text{AdaLN}(z_{t}^{k}) = (1 + \gamma_k) \cdot \text{LayerNorm}(z_{t}^{k}) + \beta_k,
> $$
> where $\gamma_k \in \mathbb{R}^C$ and $\beta_k \in \mathbb{R}^C$ are adaptive channel-wise scaling and shifting parameters, respectively. Applied to the DiT features $z_{t}^{k} \in \mathbb{R}^{C \times H \times W}$, this modulation allows AdaLN to **adaptively identify and suppress** dimensions with massive activation through the factor $(1 + \gamma_k)$. Consequently, the massive activations are effectively mitigated by the channel-wise modulation of AdaLN (see Figure 5).
>
> ### **Q2. Explanations of inconsistent dimensions in flux**
>
> Thank you for your comment. We emphasize that AdaLN effectively suppresses the original massive activations at dimensions 154 and 1446 through channel-wise modulation. The observed inconsistency in the dimensions of (weakened) massive activations stems from the adaptive nature of AdaLN, which applies distinct scaling factors to each dimension based on learned statistics. While AdaLN strongly attenuates the dimensions corresponding to massive activations, it may slightly amplify others, resulting in a shift in the top activated dimensions after normalization. This behavior reflects the rebalancing effect of AdaLN and does not negatively impact downstream performance.
>
>
> ### **Q3. Comparisons between models with and without the DiTF framework**
>
> We have provided detailed direct comparisons between the same base models (e.g., SD3-5 or Flux) with and without the DiTF framework in Figure 6. For better readability, we summarized these results in Table i. The comparison clearly shows that incorporating DiTF leads to consistent and significant performance improvements across different DiT-based models.
>
> | Model| Pixart-alpha |SD3|SD3-5|Flux|
> | :--------: | :-------: |:-------: |:-------: |:-------: |
> | w/o DiTF|35.6|24.8|39.8|29.9|
> | w DiTF |58.5|58.0|64.6|67.1|
>
> *Table i. Comparisons between models with and without the DiTF on the SPair-71k semantic correspondence benchmark.*
>
> In the paper, SD3 refers to the SD3-Medium model, while SD3-5 corresponds to the SD3.5-Large model. Additionally, we conducted experiments on SD3.5-Medium, where our DiTF led to a substantial performance improvement from **38.8 to 58.3**, demonstrating the effectiveness of our method. To improve clarity, we will provide more detailed descriptions of the different DiT variants in the final revision.
>
> ### **Q4. Generalizability on various CV tasks**
>
> We have conducted **semantic segmentation** experiments on ADE20K, as presented in Table 6 of the main paper. Specifically, our method significantly improves the performance of DiT features, increasing the mIoU **from 43.8 to 53.6** (*DiTF$_{Flux}$*). Additionally, we perform **linear probing** experiments on **ImageNet classification** using DiT features, following the same protocol as DINOv2 [1]. We use the globally pooled DiT features as input. Our method again yields a substantial improvement, boosting accuracy **from 65.8 to 72.6** (*DiTF$_{Flux}$*). These results demonstrate the effectiveness and general applicability of our approach across diverse vision tasks.
>
> ### **Q5. Discussion of limitations and future works**
>
> We have discussed the limitations and potential future directions in the supplementary material. In the final revision, we will expand upon these points and incorporate them into the main text to enhance clarity and completeness. In response to your suggestion, we will also apply color coding to Figures 1a and 2 for improved readability.
>
>
> [1] DINOv2: Learning Robust Visual Features without Supervision, TMLR 2025.

---

> > ### Comment · Reviewer_f4nx · 2025-08-06
> > **Convincing new results and explanation**
> >
> > Thanks a lot for the detailed explanation. I'm convinced of the new results and explanation that the authors provided in the rebuttal. Thus, I would like to raise my rating.

---

### Official Review · Reviewer_ntGm · 2025-07-05

**Clarity:** 3
**Significance:** 3
**Originality:** 3
**Rating:** 4
**Confidence:** 5

**Summary:**

This paper investigates the massive activation phenomenon in Diffusion Transformers (DiTs) for visual correspondence tasks. The authors identify that DiTs exhibit massive activations concentrated in specific feature dimensions across all image patch tokens, which leads to uninformative representations and performance degradation. They trace this phenomenon to the AdaLN-zero layers and propose DiTF (Diffusion Transformer Feature), a training-free framework that leverages AdaLN's channel-wise modulation to suppress massive activations and extract semantically discriminative features. The method achieves state-of-the-art performance on multiple correspondence benchmarks, with significant improvements over existing approaches.

**Questions:**

1. **Theoretical Foundation**: Can you provide theoretical insights into why massive activations appear uniformly across all image patch tokens in DiTs, unlike the token-specific patterns in LLMs? What architectural or training differences in DiTs might explain this phenomenon?
2. **Computational Efficiency**: Could you provide a detailed computational cost comparison between your billion-parameter DiT models and much smaller supervised baselines? How do you justify the computational overhead given the dramatic parameter size differences?
3. **Hyperparameter Sensitivity**: How sensitive is the method to the channel discard threshold and other hyperparameters? Can you provide ablation studies showing performance across different threshold values and discuss the method's robustness?

**Ethical Concerns:**

["NO or VERY MINOR ethics concerns only"]

**Final Justification:**

I hope that the revisions mentioned by the authors will be made as requested, and I will maintain my score as it is.

**Limitations:**

The authors should more transparently discuss several limitations in the "main" paper (not only in supplementary): (1) the lack of theoretical understanding for the unique spatial distribution of massive activations in DiTs, (2) the significant computational overhead compared to smaller supervised methods, (3) the need for hyperparameter tuning and its robustness implications, and (4) the method's reliance on pre-trained models which may limit applicability to novel architectures or training paradigms.

**Paper Formatting Concerns:**

No major formatting issues were identified. The paper follows standard academic formatting conventions with appropriate use of figures, tables, and citations.

**Quality:**

3

**Strengths And Weaknesses:**

**Strengths:**

1. **Novel Discovery and Analysis**: This work provides the first systematic analysis of massive activations in DiTs, identifying unique characteristics distinct from LLMs (concentration across all spatial locations vs. specific tokens). The connection between massive activations and AdaLN-zero residual scaling factors is insightful.
2. **Practical Training-Free Solution**: The proposed DiTF framework is elegant and immediately applicable to existing pre-trained DiTs without requiring additional training, making it highly practical for the community.
3. **Comprehensive Experimental Validation**: The authors demonstrate consistent improvements across multiple DiT architectures (Pixart-alpha, SD3, SD3-5, Flux) and datasets (SPair-71k, AP-10K, PF-Pascal), establishing strong empirical evidence.
4. **Strong Performance Gains**: The method achieves substantial improvements (+9.4% on SPair-71k, +4.4% on AP-10K-C.S.) and establishes new state-of-the-art results for DiT-based visual correspondence.

**Weaknesses:**

1. **Insufficient Theoretical Analysis**: While the paper identifies the connection between massive activations and AdaLN-zero, it lacks theoretical explanation for why DiTs exhibit massive activations across *all* image patch tokens, unlike the token-specific patterns observed in LLMs. This fundamental difference deserves deeper theoretical investigation.
2. **Inadequate Computational Cost Analysis**: The paper compares against models with vastly different computational requirements without transparent discussion. For instance, Flux operates at billion-parameter scale while supervised methods like CATs++ use ~50M parameters. While supervised methods require labeled data training, their dramatically smaller size represents a significant efficiency advantage that should be explicitly acknowledged and discussed.
3. **Limited Hyperparameter Analysis**: The paper introduces several hyperparameters (channel discard thresholds, condition dependencies) but provides insufficient sensitivity analysis. The robustness of the method across different threshold values and its stability across diverse scenarios remain unclear.
4. **Missing Related Work**: The paper fails to cite "TinyFusion: Diffusion Transformers Learned Shallow" (Fang et al., 2024), which also identified massive activation issues in DiTs and proposed Masked RepKD as a solution. This is a critical omission as it represents the first work to address massive activations in DiTs, even if at a preliminary level.
5. **Lack of Transparency in Limitations**: While the supplementary material includes a limitations section, the main paper does not transparently discuss these constraints, which is important for reader understanding and future research directions.

---

> ### Author Rebuttal · Authors · 2025-07-30
>
> Thanks for your valuable suggestions. We’re glad that you found our work insightful and interesting. Please see the responses to your comments.
>
> ### **Q1. More theoretical insights into the unique characteristics of massive activations in DiTs**
>
> Thank you for your insightful comment. We would like to emphasize that our primary focus in this work is to investigate the impact of massive activations from a representation learning perspective. To inspire further research on visual generation, we provide a preliminary theoretical claim to offer insights into the underlying cause of this phenomenon.
>
> **Distinct training paradigms and supervisory signals result in different spatial locations of massive activations in LLMs and DiTs.**
> - LLMs are typically trained using an **autoregressive next-token prediction** objective, where each token prediction is conditioned on all preceding tokens. As the model generates the entire sequence token-by-token, the early tokens play a **disproportionately important role** in shaping the final output distribution. Consequently, the initial tokens (particularly starting token) tend to accumulate stronger activations, resulting in token-specific massive activations [1].
> - In contrast, Diffusion Transformers (DiTs) are trained under a **denoising diffusion** paradigm, where supervision is applied uniformly across all image patches. Each patch is **treated equivalently** during training, and the model learns to reconstruct all patches simultaneously. This **uniform treatment** results in a spatially uniform distribution of massive activations across tokens.
> - To verify our hypothesis, we conduct a comparative analysis of massive activations in two large language models **LLaMA3-8B** [4] and **LLaDA-8B** [5]. Notably, these two models share a similar architecture, and the primary difference lies in their training paradigms: LLaMA3-8B follows **next-token prediction** objective, while LLaDA-8B is trained using **masked diffusion modeling**. As shown in Table i, massive activations in LLaMA3-8B are concentrated in a **specific token**: the starting token (<|startoftext|>). In contrast, LLaDA-8B exhibits massive activations in dimension 3848 **across all text tokens**. This distinction aligns with our theoretical claim that the choice of training paradigm fundamentally influences the activation distribution patterns.
>
>
> | Text tokens |<\|startoftext\|>|"Summer"|$~~$"is" |$~~$"warm"|$~~$"."|"Winter"|$~~$"is"|$~~$"cold"|$~~$"."|
> | :--: | :--: |:--: |:--: |:--: |:--: |:--: |:--: |:--: |:--: |
> | LLaMA3-8B (788) | 226.0|2.0|1.4|0.9|1.2|0.2|1.4|0.7|2.3|
> | LLaDA-8B (3848)|134.0|112.5|101.0|103.5|108.5|116.5|80.0|119.5|92.5|
>
> *Table i. Distribution of activation values along the massive activation dimension in LLaMA3-8B and LLaDA-8B. The dimension associated with massive activations is indicated by (·). The input sentence is: "Summer is warm. Winter is cold.".*
>
> ### **Q2. Computational cost analysis**
>
> To provide a more transparent comparison, we have included a detailed computational cost analysis in Table ii. All results are measured on a single NVIDIA L40S GPU (48 GB). Below, we elaborate on several key aspects relevant to this concern:
>
> **1) Training-free and zero-shot Inference:**
>
> DiTF operates in a fully training-free and zero-shot manner, leveraging pretrained Diffusion Transformers (DiTs) as universal feature extractors. This design enables direct application to a wide range of visual correspondence tasks **without any task-specific fine-tuning**. In contrast, supervised methods such as CATs++ [2] rely on costly and time-consuming task-specific training.
>
> **2) Efficient inference with simpler matching pipeline:**
>
> DiTF achieves faster inference despite using larger backbones, due to its **lightweight and direct matching pipeline**. Specifically, DiTF only requires a single forward pass to extract dense features from each image, followed by a **cosine similarity computation** for pixel-wise matching, where no trainable modules are introduced in the matching process. In contrast, supervised methods typically adopt a two-stage process:(1) Extract features via a CNN or Transformer backbone (2) Feed the feature pair into an **additional trainable cost aggregation network** to produce the final matching result. This additional network significantly **slows down inference**. As a result, although DiTF uses a large pretrained backbone, its inference time is **faster** than that of supervised as CATs++ [2] (**42.0 ms vs. 55.9 ms**), as shown in Table ii.
>
> **3) Negligible overhead from large-scale backbones:**
>
> The feature extraction stage consumes **only a small portion of the overall runtime**. For instance, with the billion-parameter DiT (Flux) model, the feature extraction time is 6.21ms (42.0ms in total), accounting for **less than 15%** of total inference time. This suggests that the performance gain from better features outweighs the minor cost introduced by a larger backbone.
>
> | Model |Zero-shot|Inference Time \[ms/pair\] | Accuracy |
> | :-----: | :-------: |:-------: |:-------: |
> | CATs++| No |55.9(1.99)|59.8|
> | DINOv2 |Yes|21.0(2.53)|55.6|
> | DIFT|Yes|26.7(4.08)|57.7|
> | DiTF$_{SD3-5}$|Yes|29.5(4.90)|64.6|
> | DiTF$_{Flux}$|Yes|42.0(6.21)|67.1|
>
> *Table ii. Computational cost comparison of different models on the SPair-71k semantic correspondence benchmark. Inference time for feature extraction is denoted by (·)*
>
> ### **Q3. Hyperparameter analysis**
>
> **Channel discard thresholds**: In our study, we set the default discard threshold to $m=10$, where a feature dimension is discarded (zeroed) if its mean activation magnitude **exceeds $m\times$the global median**. To assess robustness, we evaluated thresholds $m\in$ {14,12,10,8,6}. As Table iii shows, our method maintains stable performance across all thresholds and different DiT models. Notably, the normalized (weakened) massive activations remain ≥14× the global median, ensuring a clear distinction from normal activations. Such a wide margin ensures that varying the threshold $m$ (from 6 to 14) has minimal impact on performance, highlighting the robustness of our method.
>
> | Threshold  | $~$14 | $~$12 | $~$10 | $~~$8 | $~$6 |
> | :------------: | :--: | :--: | :--: | :--: | :--: |
> | Pixart-Alpha   | 58.5 | 58.5 | 58.5 | 58.5 | 58.5 |
> | SD3-5          | 64.6 | 64.6 | 64.6 | 64.6 | 64.6 |
> | Flux           | 67.1 | 67.1 | 67.1 | 67.1 | 66.5 |
>
> *Table iii. Performance with different channel discard thresholds on the SPair-71k semantic correspondence benchmark.*
>
> **Condition dependencies**: We have already included a detailed ablation study in Table 5, demonstrating that the condition timestep t plays a critical role in enabling AdaLN to modulate massive activations (Lines 248-254). More investigations of the block index k and the timestep t are provided in Figure 4 of the supplementary material.
>
> ### **Q4. Related works**
> We appreciate the suggestion and will cite TinyFusion [3] along with other related works in the revised version.
>
> ### **Q5. Transparent discussion of limitations**
> In the revised version, we will move the limitations and future work section from the supplementary material to the main paper, and include a discussion emphasizing the need for a deeper understanding of why such massive activations emerge during DiT training and what functional role they play in visual generation tasks.
>
> [1] Massive activations in large language models, COLM 2024.
>
> [2] CATs++: Boosting Cost Aggregation with Convolutions and Transformers, TPAMI 2022.
>
> [3] TinyFusion: Diffusion Transformers Learned Shallow, CVPR2025.
>
> [4] The llama 3 herd of models.
>
> [5] Large Language Diffusion Models.

---

> > ### Comment · Reviewer_ntGm · 2025-08-07
> >
> > ## **Response to Authors' Rebuttal**
> >
> > Thank you for your comprehensive rebuttal addressing most of my concerns. While I appreciate the theoretical insights on training paradigms (Q1), computational cost analysis (Q2), and hyperparameter sensitivity analysis (Q3), I must express significant concern regarding your response to the TinyFusion citation issue (Q4).
> >
> > **Critical Issue: Misrepresentation of Main Contribution**
> >
> > Your response to Q4 simply states "We appreciate the suggestion and will cite TinyFusion [3] along with other related works in the revised version." This dismissive treatment of a fundamental issue is **unacceptable** and represents a serious mishandling of your claimed contributions.
> >
> > **The Problem:**
> > - Your main contribution explicitly claims: *"We are the first to identify and characterize massive activations in Diffusion Transformers (DiTs)"* (Lines 55-57)
> > - TinyFusion (Fang et al., CVPR 2025) **already identified and characterized** massive activations in DiTs, proposing Masked RepKD as a solution
> > - Your response treats this as a minor citation oversight rather than addressing the fundamental invalidation of your primary novelty claim
> >
> > **Required Action:**
> > This is not merely a citation issue—it requires **substantial revision of your main contribution claims**. You cannot simply add TinyFusion to related work while maintaining your "first to identify" claim. The contribution statement must be fundamentally reframed to:
> >
> > 1. **Acknowledge TinyFusion's prior identification** of massive activations in DiTs
> > 2. **Clearly differentiate your contributions** from theirs (e.g., systematic analysis, AdaLN-based solution, theoretical insights)
> > 3. **Rewrite the main contribution bullets** to reflect the actual novel aspects of your work
> > 4. **Provide proper positioning** in the introduction and related work sections
> >
> > **Academic Integrity Concern:**
> > Treating this as a simple citation addition while maintaining false priority claims raises serious questions about academic integrity. The community relies on accurate representation of contributions, and your current approach undermines this trust.
> >
> > **Conditional Assessment:**
> > I want to emphasize that your technical work is solid and makes valuable contributions to the field. However, the misrepresentation of novelty is a critical flaw that must be addressed comprehensively.
> >
> > **If the revised paper properly acknowledges prior work and accurately frames contributions**, I maintain my borderline accept assessment. **However, if you proceed with only superficial citation additions while maintaining false priority claims**, I will strongly recommend rejection based on misrepresentation of contributions.
> >
> > **Specific Revision Requirements:**
> > 1. Rewrite Lines 55-57 and all related contribution claims
> > 2. Add comprehensive discussion of TinyFusion in introduction and related work
> > 3. Clearly position your work as advancing/extending prior identification rather than first discovery
> > 4. Ensure abstract and conclusion accurately reflect the positioned contributions
> >
> > I urge you to take this feedback seriously and provide a substantial revision that maintains academic integrity while properly crediting prior work.

---

> > > ### Author Response · Authors · 2025-08-07
> > >
> > > Thank you for your constructive and detailed feedback. We truly appreciate the time and effort you put into evaluating our work.
> > >
> > > We sincerely apologize for the imprecise claim in our original phrasing regarding being the “first” to identify and characterize massive activations in DiTs. When we initially formulated this idea and began our investigation, it was in late 2024. During the development process, we carelessly overlooked the recently published TinyFusion [1], which already identified and characterized massive activations in DiTs.
> > >
> > > This was an **honest oversight rather than an intentional misrepresentation**. We fully agree with your assessment that this is not merely a citation issue but a matter of properly framing the novelty of our contributions. We take academic integrity very seriously.
> > >
> > > To address this concern thoroughly, we will **comprehensively revise the contribution statements, introduction, and related work sections**, and ensure that all other parts of the paper **accurately reflect** the properly positioned contributions.
> > >
> > > ### **Contribution claims**
> > > We **fully acknowledge that TinyFusion [1] has previously identified the massive activations** in DiTs. In the final revision, we will carefully revise and clarify our contribution statements (Lines 55-57) to distinguish our work. Specifically, we highlight the following key contributions:
> > >
> > > - *We conduct a comprehensive and systematic analysis of massive activations in DiTs and find them consistently concentrated in a small number of fixed feature dimensions across all image patch tokens.*
> > > - *We trace the source of these dimensionally concentrated massive activations and reveal their strong connection to the AdaLN-Zero layer in DiTs. Furthermore, our analysis suggests that the spatial patterns of massive activations are likely influenced by the training paradigms adopted in models.*
> > > - *We propose a training-free, AdaLN-based framework, Diffusion Transformer Feature (DiTF), for extracting semantically discriminative representations from DiTs.*
> > > - *Our framework outperforms both DINO and SD-based models, establishing a new state-of-the-art for Diffusion Transformers on visual correspondence tasks.*
> > >
> > >
> > > ### **Comprehensive discussion of TinyFusion in introduction and related work**
> > > Thank you for your valuable suggestions. We will comprehensively discuss TinyFusion and our work in the Introduction and Related Work sections to clearly and accurately clarify our contributions. Specifically:
> > >
> > > - *"Recent work by TinyFusion [1] has identifies massive activation outliers in Diffusion Transformers (DiTs), which cause unstable training and excessively high distillation losses during knowledge distillation. To address this, TinyFusion proposes Masked RepKD, a method that suppresses outlier activations via thresholding while preserving normal knowledge transfer.*
> > > - *In this work, we advance prior identification of massive activations in DiTs by conducting a comprehensive and systematic analysis to better understand their nature and origin. We find that these massive activations are consistently concentrated in a small number of fixed feature dimensions across all image patch tokens. Furthermore, our investigation uncovers a strong connection between this dimensional concentration and the AdaLN-Zero layer, and shows that the training paradigms of models induce spatially uniform massive activation patterns. Building on these insights, we propose a training-free, AdaLN-based feature extraction framework that substantially improves the performance of pretrained DiTs on dense prediction tasks."*
> > >
> > > In summary, while TinyFusion identifies the existence of massive activations in DiTs, our work **extends and advances this line of study by systematically analyzing** both their dimensional patterns and spatial distributions. We further provide **novel insights** into the origins of these unique activation characteristics, revealing their strong connection to AdaLN-zero layer in DiTs and training paradigms of models.
> > >
> > > [1] Massive activations in large language models, COLM 2024.

---

> > > > ### Author Response · Authors · 2025-08-07
> > > >
> > > > ### **Clearly position our work as advancing/extending prior identification rather than first discovery**
> > > > In the final revision, we will carefully and clearly position our work as an extension of prior identification rather than a first discovery. Specifically:
> > > >
> > > > - **TinyFusion [1] has identified** the presence of massive activations in DiTs and addressed this issue through Masked RepKD for lightweight model design. In contrast, our work **advances this prior identification by presenting a substantially deeper analysis** of massive activations in DiTs. Specifically, we investigate both their **dimensional patterns** and **spatial distributions** to better understand their underlying characteristics. We find that these activations are consistently concentrated in a small number of fixed feature dimensions and uncover a **strong connection to the AdaLN-Zero layer**. Moreover, our analysis suggests that the spatially uniform distribution of massive activations is likely influenced by the **training paradigms** adopted in models.
> > > >
> > > >
> > > > ### **Ensure abstract and conclusion accurately reflect the positioned contributions**
> > > > Thank you for your valuable suggestion. To ensure academic rigor and clarity, we will carefully revise all parts of the paper (include abstract and conclusion) to accurately position our contributions as **advancing and extending prior work** rather than claiming first discovery. Specifically, the abstract will be updated to emphasize our deeper, systematic analysis of massive activations, as well as our novel AdaLN-based feature extraction framework that significantly enhances pretrained DiTs. Similarly, the conclusion will be rewritten to reflect these points consistently.
> > > >
> > > > For example, the original conclusion:
> > > >
> > > > - *"In this paper, we identify and characterize massive activations in Diffusion Transformers (DiTs). We observe that these massive activations are concentrated in a few fixed dimensions across all image patch tokens and carry limited local information."*
> > > >
> > > > will be revised to:
> > > >
> > > > - *"In this paper, we build upon prior identification and provide a more comprehensive and in-depth analysis for the massive activations in DiTs. We observe that these activations are consistently concentrated in a few fixed dimensions across all image patch tokens and contain limited local information."*
> > > >
> > > > Throughout the manuscript, we will maintain this clear and accurate representation to provide readers with a precise understanding of our contributions.
> > > >
> > > >
> > > > [1] Massive activations in large language models, COLM 2024.

---

> > > > > ### Comment · Reviewer_ntGm · 2025-08-07
> > > > >
> > > > > I hope that the revisions mentioned by the authors will be made as requested, and I will maintain my score as it is.

---

> > > > > > ### Author Response · Authors · 2025-08-07
> > > > > >
> > > > > > Thank you for your continued recognition and support. We would like to reaffirm that we will make the revisions as requested in the final version to ensure clarity and completeness. Your feedback has been very helpful in improving the quality of our work.

---

### Comment · Area_Chair_BsEN · 2025-08-04

Dear Reviewers,

As we near the end of the rebuttal period, this is a friendly reminder to submit your responses to the authors by **August 6**. Your engagement is crucial for our final decision.

**Action Required**

- Read the rebuttal carefully: Authors have invested significant effort in addressing your concerns.

- Reply directly to authors: Briefly acknowledge their points and indicate whether your assessment has changed (or why it remains unchanged).

- Update your review (if applicable): Adjust scores/comments in the review system to reflect your current stance.

Your AC

---

### Decision · Program_Chairs · 2025-09-17

**Decision:**

Accept (poster)

**Comment:**

This paper investigates Diffusion Transformers (DiTs) for dense visual correspondence and highlights the phenomenon of massive activations, where a small number of feature activations exhibit disproportionately large values. Such activations degrade feature quality and hinder performance. To address this issue, the authors introduce Diffusion Transformer Feature (DiTF), a training-free framework that employs AdaLN modulation and channel discard to suppress these activations.

The strengths of the work lie in its systematic analysis of massive activations, its practical and training-free solution, and its comprehensive experimental validation across diverse architectures and datasets. Reviewers emphasized that the framework is elegant, readily applicable to existing pretrained DiTs, and delivers significant performance improvements.

After the rebuttal, most concerns were satisfactorily resolved. The authors provided theoretical insights into activation differences among LLMs, ViTs, and DiTs, clarified the computational efficiency of their method, and demonstrated robustness across hyperparameter settings. They also acknowledged related work (TinyFusion) and committed to revising their novelty claims, while presenting additional experiments on segmentation and classification to support generalization further.

In conclusion, this paper offers a solid contribution backed by substantial empirical evidence. I recommend acceptance and encourage the authors to incorporate the reviewers’ suggested revisions into the final version.